# The middle lipin domain adopts a membrane-binding dimeric protein fold

Weijing Gu [1,4], Shujuan Gao[1,4], Huan Wang [2,4], Kaelin D. Fleming [3], Reece M. Hoffmann[3], Jong Won Yang[1], Nimi M. Patel [1], Yong Mi Choi[1], John E. Burke [3,5], Karen Reue [2,5✉] & Michael V. Airola [1,5✉]

Phospholipid synthesis and fat storage as triglycerides are regulated by lipin phosphatidic acid phosphatases (PAPs), whose enzymatic PAP function requires association with cellular membranes. Using hydrogen deuterium exchange mass spectrometry, we find mouse lipin 1 binds membranes through an N-terminal amphipathic helix, the Ig-like domain and HAD phosphatase catalytic core, and a middle lipin (M-Lip) domain that is conserved in mammalian and mammalian-like lipins. Crystal structures of the M-Lip domain reveal a previously unrecognized protein fold that dimerizes. The isolated M-Lip domain binds membranes both in vitro and in cells through conserved basic and hydrophobic residues. Deletion of the M-Lip domain in lipin 1 reduces PAP activity, membrane association, and oligomerization, alters subcellular localization, diminishes acceleration of adipocyte differentiation, but does not affect transcriptional co-activation. This establishes the M-Lip domain as a dimeric protein fold that binds membranes and is critical for full functionality of mammalian lipins.

[1] Department of Biochemistry and Cell Biology, Stony Brook University, Stony Brook, NY, USA. [2] Department of Human Genetics, David Geffen School of Medicine at UCLA, Los Angeles, CA, USA. [3] Department of Biochemistry and Microbiology, University of Victoria, Victoria, BC, Canada. [4] These authors contributed equally: Weijing Gu, Shujuan Gao, Huan Wang. [5] These authors jointly supervised this work: John E. Burke, Karen Reue, Michael V. Airola. ✉email: reuek@ucla.edu; michael.airola@stonybrook.edu

ipins are magnesium-dependent phosphatidic acid phos-
phatases (PAPs) that catalyze the dephosphorylation of the
membrane lipid phosphatidic acid (PA) to produce dia-
cylglycerol (DAG)[1]. The conversion of PA to DAG by lipins
regulates de novo phospholipid biosynthesis[2], cellular signaling[3],
chylomicron biogenesis[4], adipocyte differentiation[5], and fat sto-
rage as triglycerides[6]. Mutations that reduce lipin PAP activity[7–9]
are associated with metabolic diseases including
rhabdomyolysis[10], Majeed syndrome[11], lipodystrophy[6], statin-
induced myopathy[12,13], and insulin resistance[6,14]. Mammalian
lipins also function as transcriptional co-activators to affect the
transcription of genes for fatty acid oxidation[8,15]. There are three
mammalian lipin paralogs: lipin 1, lipin 2, and lipin 3[6,16]. Many
insights into lipin PAP function have come from studies of the
yeast lipin homolog, Saccharomyces cerevisiae PA phosphohy-
drolase 1 (Sc Pah1)[1,17].

The architecture of mammalian lipins and Sc Pah1 differ, but
all lipin/Pah homologs share two common and conserved regions
called the N-Lip and C-Lip regions, which are located at the
respective N- and C-termini of mammalian lipins (Fig. 1a)[6].
Recently, we determined the structure of Tetrahymena thermo-
phila Pah2 (Tt Pah2), which revealed the N-Lip and C-Lip
regions co-fold to form a catalytic unit comprised of a split
immunoglobulin-like (Ig-like) domain and a haloacid
dehalogenase-like (HAD-like) catalytic domain[18]. The N-Lip
and C-Lip regions are separated by an extended linker that varies
in length and sequence across species[17,19]. In mammalian
lipins, this linker is 500 amino acids and can be
hyperphosphorylated[7,20–22], sumoylated[23], or acetylated[24].
Phosphorylation, sumoylation, or acetylation within the linker
region is reported to regulate the subcellular location and activity
of mammalian lipins[7,21–25]. However, it is not known if the linker
region has additional roles.

As the only enzyme in the glycerol-3-phosphate pathway that
is not constitutively membrane-bound, the regulation of lipin/Pah
membrane association is a determinant of its enzyme activity. In
vitro, purified Sc Pah1, Tt Pah2, and mammalian lipins are
recruited to membranes containing their substrate, PA[18,22,26].
Lipin/Pahs lack canonical lipid binding domains (e.g., PH and PX
domains) found in other membrane-binding proteins, but con-
tain a conserved N-terminal amphipathic helix that is necessary
for lipin/Pahs to bind membranes in vitro and in cells[18,26] In
addition, a nuclear localization signal/polybasic region in mam-
malian lipins has been implicated in membrane binding[27,28].

We sought to characterize how lipins bind membranes and in
the process identified a new middle lipin (M-Lip) domain that is
universally conserved in mammalian and mammalian-like lipins,
but not present in Sc Pah1. Herein, we report the structural and
functional characterization of the M-Lip domain. Our principal
findings are that the M-Lip domain is a new protein fold that
forms a dimer, binds membranes, and can affect lipin PAP
activity, oligomerization, subcellular localization, and
adipogenesis.

## Results

**Structure and dynamics of lipin 1**. Full-length mouse lipin 1α
(herein referred to as lipin 1) was purified from Sf9 cells and the
structure and dynamics were probed using hydrogen deuterium
exchange mass spectrometry (HDX-MS). HDX-MS measures the
exchange rate of amide hydrogens with deuterium. The major
determinant of exchange is the stability of secondary structure[29].
Thus, HDX-MS provides a readout of secondary structure
dynamics. In addition, HDX experiments with extremely short
exposures of $D_2O$ can be used to identify disordered regions
within proteins when compared to a fully deuterated condition[29].

HDX experiments were carried out with a short pulse of
deuterium exposure to map regions of order/disorder in full-
length lipin 1. A total of 189 peptides spanning 83.6% of the
primary sequence were identified and their deuterium content
was quantified. Peptides for residues 121-256 and residue 413
were not identified by tandem MS/MS. Residues with no MS
coverage were all located between the N-Lip and C-Lip regions,
and near the highly basic nuclear localization signal.

The N-Lip and C-Lip regions both had low rates of deuterium
exchange with a 3 s pulse of $D_2O$ exposure at 4 °C, which
indicates these regions were largely ordered into secondary
structure elements (Fig. 1b). This suggests that the N-Lip and C-
Lip regions co-fold to form the split Ig-like domain and HAD-like
domain observed in the Tt Pah2 structure[18], and that in vitro
their interaction is most likely constitutive.

The majority of the >500 residues that separate the N-Lip and
C-Lip regions had high rates of deuterium exchange, indicating
that they are likely disordered (Fig. 1b). One exception was a
continuous stretch of ~100 residues that were protected from
exchange, which is indicative of secondary structure formation
(Fig. 1b). This ordered region is called the middle lipin (M-Lip)
domain and we discuss M-Lip in more detail below.

**Lipin 1 association with membranes**. In line with previous
observations[22,27], recombinant lipin 1 bound strongly to PC/PA
liposomes (see below). To identify the membrane-binding regions
of lipin 1 we employed HDX-MS, which has been particularly
useful in examining protein-lipid interactions[30,31]. HDX-MS
experiments were carried out in the presence and absence of PC/
PA liposomes and at 4 different time points of exchange (3, 30,
300, and 3000 s).

HDX-MS revealed multiple peptides that were protected from
H/D exchange in the presence of liposomes, which suggests these
regions associate with membranes. The regions protected by
membrane were distributed throughout the primary structure and
clustered into several key areas (Fig. 1c–e, Supplementary Fig. 1a,
c). This included (i) the N-terminus (residues 2–12, 13–30) that is
predicted to form an amphipathic helix (Fig. 1c) as in other
PAPs[18,26], (ii) the C-terminal end of the M-Lip domain (residues
544–555) that is enriched in basic and hydrophobic residues
(Fig. 2a), (iii) peptides in the Ig-like domain (residues 651–662)
that are predicted to lie at the membrane interface, (iv) the
catalytic active site of the HAD-like domain (residues 660–685,
724–732) where PA hydrolysis occurs, and (v) the C-terminal end
of the C-Lip (residues 840–845, 855–867) that is situated between
the end of the HAD-like domain and the conserved Trp motif[32]
(Fig. 1a–e, Supplementary Fig. 1a, c). As stated above, peptides
containing the NLS were not identified by tandem MS/MS. Thus,
we were unable to assess the dynamics of membrane association
for the NLS, which has previously been implicated in lipin 1
membrane binding[27].

Notably, the membrane protected regions within the N-Lip and
C-Lip regions of lipin 1 (Fig. 1c, Supplementary Fig. 1a, c) were
nearly identical with those previously observed for Tt Pah2[18].
This suggests the catalytic core of PAP enzymes utilize a
conserved mechanism for membrane binding that involves an
N-terminal amphipathic helix, the HAD-like active site, and
portions of the Ig-like domain.

**The M-Lip domain**. We next turned our attention to M-Lip, as
the HDX-MS experiments suggested it may represent a third
domain in lipin 1 that is involved in membrane binding. BLAST
searches revealed that the M-Lip domain was selectively found in
lipin homologs and was absolutely conserved in all mammalian
lipins (Fig. 2a). The M-Lip domain was also detected in some

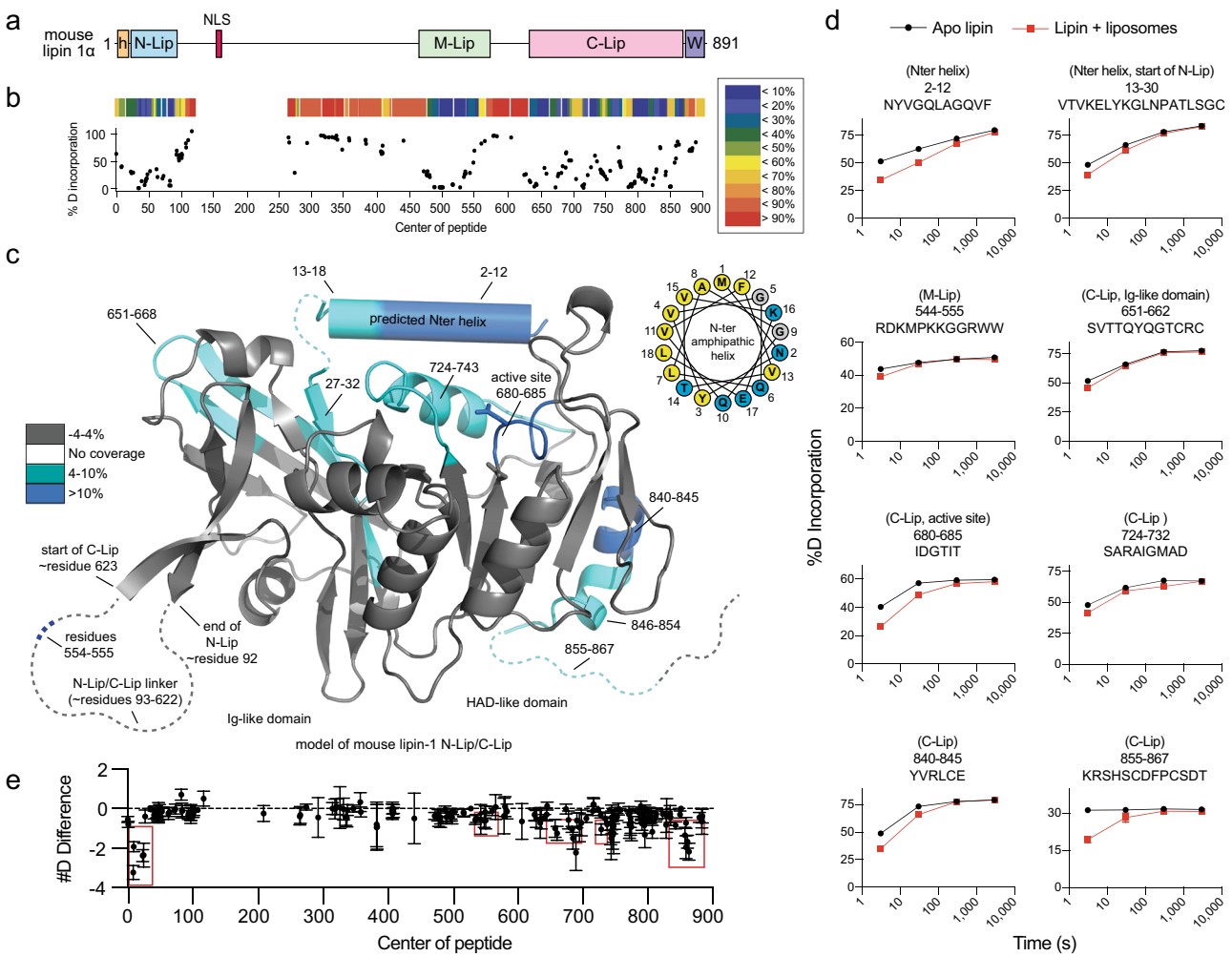

**Fig. 1 Structure and dynamics of lipin 1. a** Domain architecture of mouse lipin 1 drawn to scale. Mammalian lipins conserve three regions/domains: the N-Lip, M-Lip, and C-Lip. The N-Lip and C-Lip are predicted to co-fold to form two domains: an Ig-like and HAD-like domain. The positions of the N-ter amphipathic helix (orange h), nuclear localization signal (NLS), and conserved Trp-motif (purple W) are indicated. **b** % deuterium incorporation after a 3 s (on ice) deuterium exposure in the absence of liposomes. Each point represents a single peptide, with them being graphed on the x-axis according to their central residue. A heat map above is color coded according to the legend. **c** Regions of lipin 1 that showed significant decreases in exchange (defined as >4%, >0.4 Da, and a two-tailed student t-test $p < 0.01$) in the presence of liposomes are colored in blue according to the legend and shown on a model of the N-Lip and C-Lip regions generated by Phyre2. Regions not modeled are shown as dashed lines. Helical wheel diagram (top right) of the putative N-terminal amphipathic helix. Hydrophobic residues, yellow; polar residues, blue; glycine residues, gray. **d** % deuterium incorporation of selected peptides at various time points (3, 30, 300, and 3000 s) in the absence and presence of liposomes. Data are presented as mean values ± SDs. $n = 3$ independent experiments. Most SDs are smaller than the size of the point. **e** The sum of the # of deuterons protected from exchange in the presence of liposomes across all timepoints is shown. Each point represents a single peptide, with them being graphed on the x-axis according to their central residue. Data are presented as mean values ± SDs. $n = 3$ independent experiments. Source data are provided as a Source data file.

plant (*A. thaliana*), fungal (*Cryptococcus neoformans*), ciliate (*Tetrahymena thermophila*), insect (*Drosophila melanogaster*), and apicomplexan (*Plasmodium falciparum*) lipin homologs (Supplementary Fig. 2) but was not detected in *Sc* Pah1. The M-Lip domain is thus one feature that distinguishes mammalian and mammalian-like lipin PAPs from *Sc* Pah1.

**Structure of the M-Lip domain reveals a new protein fold.** The M-Lip domain did not share sequence homology with any domains of known function. To determine if M-Lip was indeed a protein domain with a defined tertiary structure, we sought to determine its structure. The mouse lipin 1 M-Lip domain was purified from *Escherichia coli*. The detergent Triton X-100 was required to prevent the M-Lip domain from pelleting during centrifugation after cell lysis but was not required in subsequent purification steps if high salt concentrations were maintained.

Despite extensive efforts, we have yet to successfully crystallize the complete M-Lip domain.

We, therefore, truncated the M-Lip domain of mouse lipin 1 to remove the C-terminal cluster of hydrophobic and basic residues (Fig. 2a) implicated in membrane binding by HDX-MS (Fig. 1d). We refer to this construct as the M-Lip^xtal domain. The M-Lip^xtal domain could be purified without detergent and was extremely stable with a melting temperature of 65 °C (Supplementary Fig. 3). The structure of the M-Lip^xtal domain of mouse lipin 1 was determined to resolutions of 1.5 Å and 1.9 Å in two unique space groups (Table 1, Supplementary Fig. 4a, b). Phases were obtained using single-wavelength anomalous diffraction from selenomethionine (SeMet)-derivatized protein (Table 1). We also determined the structure of the mouse lipin 2 M-Lip^xtal domain to 2.5 Å resolution (Table 1, Supplementary Fig. 4c). The M-Lip^xtal domain from mouse lipin 3 was easily purified but did not crystallize.

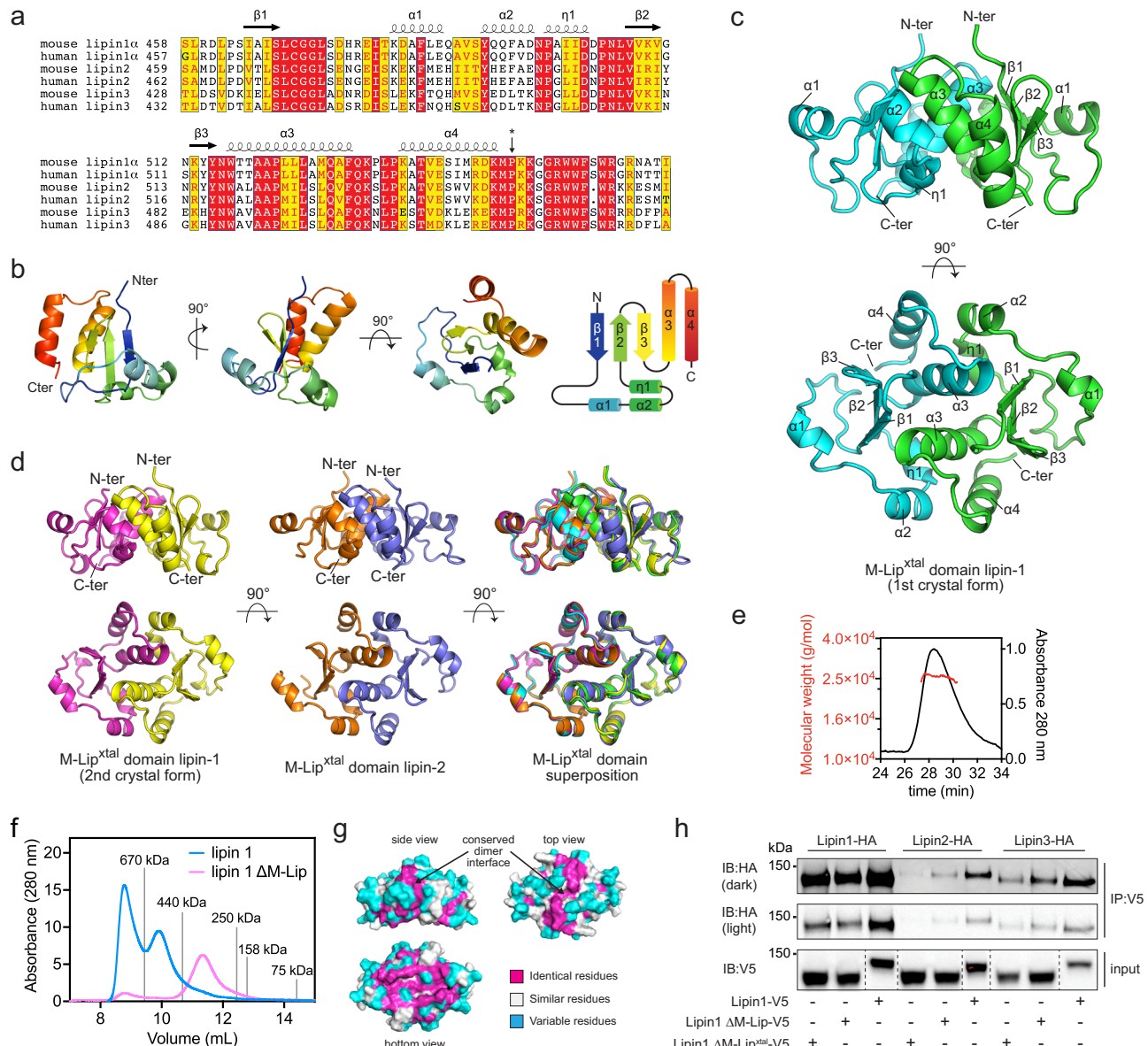

**Fig. 2 The M-Lip domain is a new protein fold that dimerizes. a** Sequence alignment of human and mouse lipin M-Lip domains with secondary structure elements above. The asterisk indicates the last residue of the M-Lip$^{xtal}$ domain. Red and yellow boxes indicate identical and positive homology residues. **b** Structure of a single subunit of the lipin 1 M-Lip$^{xtal}$ domain in three different views. The N- and C-termini are colored blue and red, respectively. **c** The M-Lip$^{xtal}$ domain forms a dimer with the α3 helices mediating the majority of contacts at the dimer interface. Individual subunits are colored green and cyan. **d** Crystal structures and superimposition of the near identical lipin 1 and lipin 2 M-Lip$^{xtal}$ domains. **e** MALS data (left axis) with SEC traces (right axis) for the lipin 1 M-Lip$^{xtal}$ domain reports a MW of 25 kDa, consistent with a dimer (MW of 26 kDa). **f** SEC profiles of full-length and lipin 1 ΔM-Lip on SEC with gray lines indicating MW standards. Deletion of the M-Lip shifts the elution profile to a smaller apparent MW. **g** Surface views of the lipin 1 M-Lip$^{xtal}$ domain showing conservation of the dimer interface. **h** Co-immunoprecipitation of HA-tagged lipins with V5-tagged lipin 1 constructs. Hepa1-6 cells were co-transfected with HA-tagged lipin 1, −2, and −3 and either V5-tagged lipin 1, ΔM-Lip, or ΔM-Lip$^{xtal}$. Deletion of the ΔM-Lip or ΔM-Lip$^{xtal}$ domains reduced the amount of HA-tagged lipin proteins that co-immunoprecipitated. $n = 1$ independent experiment. Source data are provided as a Source data file.

The structure of the M-Lip$^{xtal}$ domain revealed a protein fold with a three-stranded anti-parallel β-sheet at the core (Fig. 2b). Flanking the β-sheet were a set of two alpha helices (α1 and α2) and a short 3–10 helix (η1) that were oriented perpendicular to the β-sheet and two C-terminal α-helices (α3 and α4) that were oriented parallel to the β-sheet. A Dali search[33] did not identify any similar existing protein structures. Thus, the M-Lip domain represents a previously unrecognized and novel protein fold.

**The M-Lip domain forms a dimer.** The M-Lip domain of mouse lipin 1 and lipin 2 formed a symmetric or near symmetric dimer in all three crystal forms with a root mean square deviation for all atoms ranging between 0.36–0.77 Å (Fig. 2c, d). The observation of near identical dimers in three different crystal structures strongly suggested the native quaternary structure of the M-Lip domain as dimeric. To confirm this, we used size-exclusion chromatography coupled to multi-angle light scattering (SEC-MALS) to calculate the molecular weight (MW) of the M-Lip$^{xtal}$ domain in solution. SEC-MALS reported a MW of $25.6 ± 0.6$ kDa for the mouse lipin 1 M-Lip$^{xtal}$ domain, which was consistent with the MW of 26 kDa for a M-Lip$^{xtal}$ dimer (Fig. 2e).

**Table 1 Data collection and refinement statistics.**

| PDB ID | lipin 1 M-Lip^xtal Se-Met | lipin 1 M-Lip^xtal 7KIH | lipin 1 M-Lip^xtal crystal form 2 7KIL | lipin 2 M-Lip^xtal 7KIQ |
|---|---|---|---|---|
| Wavelength (Å) | 0.9793 | 0.9792 | 1.282 | 0.9794 |
| Resolution range (Å) | 27.27–1.96 (2.03–1.96) | 33.72–1.47 (1.52–1.47) | 33.76–1.90 (1.97–1.90) | 42.31–2.52 (2.61–2.52) |
| Space group | C 2 2 21 | C 2 2 21 | P 1 21 1 | P 1 21 1 |
| Unit cell | 41.98 59.63 67.49 90 90 90 | 44.14 59.32 67.44 90 90 90 | 34.28 58.55 42.90 90 100.057 90 | 70.71 84.61 97.51 90 108.901 90 |
| Total reflections | 105,048 (7304) | 81,785 (4098) | 88,062 (7815) | 25,1319 (24,007) |
| Unique reflections | 6250 (523) | 15,309 (1344) | 13,260 (1280) | 36,270 (3395) |
| Multiplicity | 16.8 (14.0) | 5.3 (3.0) | 6.6 (6.1) | 6.9 (7.1) |
| Completeness (%) | 94.45 (69.54) | 98.27 (87.61) | 99.08 (95.69) | 98.47 (92.83) |
| Mean I/sigma(I) | 11.62 (2.21) | 14.98 (1.32) | 3.93 (1.28) | 5.51 (1.02) |
| Wilson B-factor | 37.53 | 24.85 | 19.75 | 44.67 |
| R-merge | 0.1348 (0.6495) | 0.05211 (0.5989) | 0.2773 (0.8293) | 0.2552 (2.017) |
| R-meas | 0.1393 (0.6495) | 0.05739 (0.7061) | 0.3008 (0.9092) | 0.2761 (2.176) |
| R-pim | 0.0346 (0.164) | 0.0236 (0.364) | 0.115 (0.367) | 0.104 (0.813) |
| CC1/2 | 0.997 (0.964) | 0.996 (0.849) | 0.978 (0.627) | 0.988 (0.597) |
| CC* | 0.999 (0.991) | 0.999 (0.958) | 0.994 (0.878) | 0.997 (0.865) |
| Reflections used in refinement | | 15240 (1337) | 13181 (1264) | 36209 (3381) |
| Reflections used for R-free | | 710 (61) | 1314 (126) | 1842 (160) |
| R-work | | 0.1821 (0.4554) | 0.2221 (0.3561) | 0.2126 (0.3199) |
| R-free | | 0.2148 (0.5239) | 0.2561 (0.4178) | 0.2518 (0.3447) |
| CC (work) | | 0.978 (0.910) | 0.923 (0.767) | 0.951 (0.807) |
| CC (free) | | 0.965 (0.843) | 0.941 (0.685) | 0.948 (0.750) |
| Number of non-hydrogen atoms | | 847 | 1604 | 6814 |
| Macromolecules | | 726 | 1352 | 6550 |
| Ligands | | 0 | 19 | 10 |
| Solvent | | 121 | 233 | 254 |
| Protein residues | | 88 | 174 | 810 |
| RMS (bonds) | | 0.004 | 0.006 | 0.005 |
| RMS (angles) | | 1.08 | 1.15 | 0.65 |
| Ramachandran favored (%) | | 98.84 | 99.41 | 99.22 |
| Ramachandran allowed (%) | | 1.16 | 0.59 | 0.65 |
| Ramachandran outliers (%) | | 0.00 | 0.00 | 0.13 |
| Rotamer outliers (%) | | 0.00 | 0.00 | 0.14 |
| Clashscore | | 4.80 | 10.98 | 7.54 |
| Average B-factor | | 43.01 | 27.73 | 52.73 |
| Macromolecules | | 41.48 | 25.82 | 52.79 |
| Ligands | | n/a | 45.92 | 89.22 |
| Solvent | | 52.20 | 37.35 | 49.80 |

Statistics for the highest-resolution shell are shown in parentheses.

Mammalian lipins are known to form both homo and hetero-oligomers[34]. We hypothesized that the M-Lip domain may be involved in lipin oligomerization. We, therefore, deleted the M-Lip domain in the context of full-length mouse lipin 1 (ΔM-Lip lipin 1) and purified ΔM-Lip lipin 1 from Sf9 cells. In comparison to wild-type lipin 1, the size exclusion profile of ΔM-Lip lipin 1 was shifted towards a lower molecular weight (Fig. 2f, Supplementary Fig. 5). This is consistent with a role for the M-Lip domain in lipin 1 dimerization.

Analysis of the dimer interface of the M-Lip revealed an extensive network of interactions between the two subunits with residues within the α3 helix, which was located in the center of the dimer, mediating the majority of these interactions (Fig. 2c). Notably, the residues involved in dimerization were highly conserved among human and mouse lipin 1, lipin 2, and lipin 3 paralogs (Fig. 2a, g). Thus, we suspected that the M-Lip domain might also be involved in lipin hetero-oligomerization. Consistent with this hypothesis, deletion of the complete M-Lip domain or the M-Lip^xtal domain modestly reduced the ability of lipin 1 to co-immunoprecipitate with lipin 1, lipin 2, and lipin 3 (Fig. 2h, Supplementary Fig. 6).

**The M-Lip is not necessary for lipin transcriptional co-activator activity.** The high conservation of the M-Lip region among mammalian lipins suggested that it plays a role(s) in lipin function. Mammalian lipins function both as PAP enzymes and as transcriptional co-activators for PPARα and PGC1α with an LxxLL motif in the C-Lip critical for this latter function[15]. Since *Sc* Pah1 has neither an M-Lip domain nor transcription co-activator activity, we hypothesized M-Lip may be necessary for lipin co-activation function.

Using HEK293 cells and the established luciferase-based assay[8,15], lipin 1 increased the transcription of luciferase under the control of three peroxisome proliferator response elements (PPREs) in the presence of peroxisome-proliferator-activated receptor alpha (PPARα) and retinoic X receptor alpha (RXRα) (Fig. 3a). Transcription was further increased in the presence of peroxisome proliferator-activated receptor gamma co-activator 1-alpha (PGC1α) (Fig. 3a). Deletion of either the M-Lip or M-Lip^xtal sequences had no significant effect on PGC1α co-activation (Fig. 3a). We concluded that the M-Lip domain is not necessary for lipin 1 to function as a transcriptional co-activator.

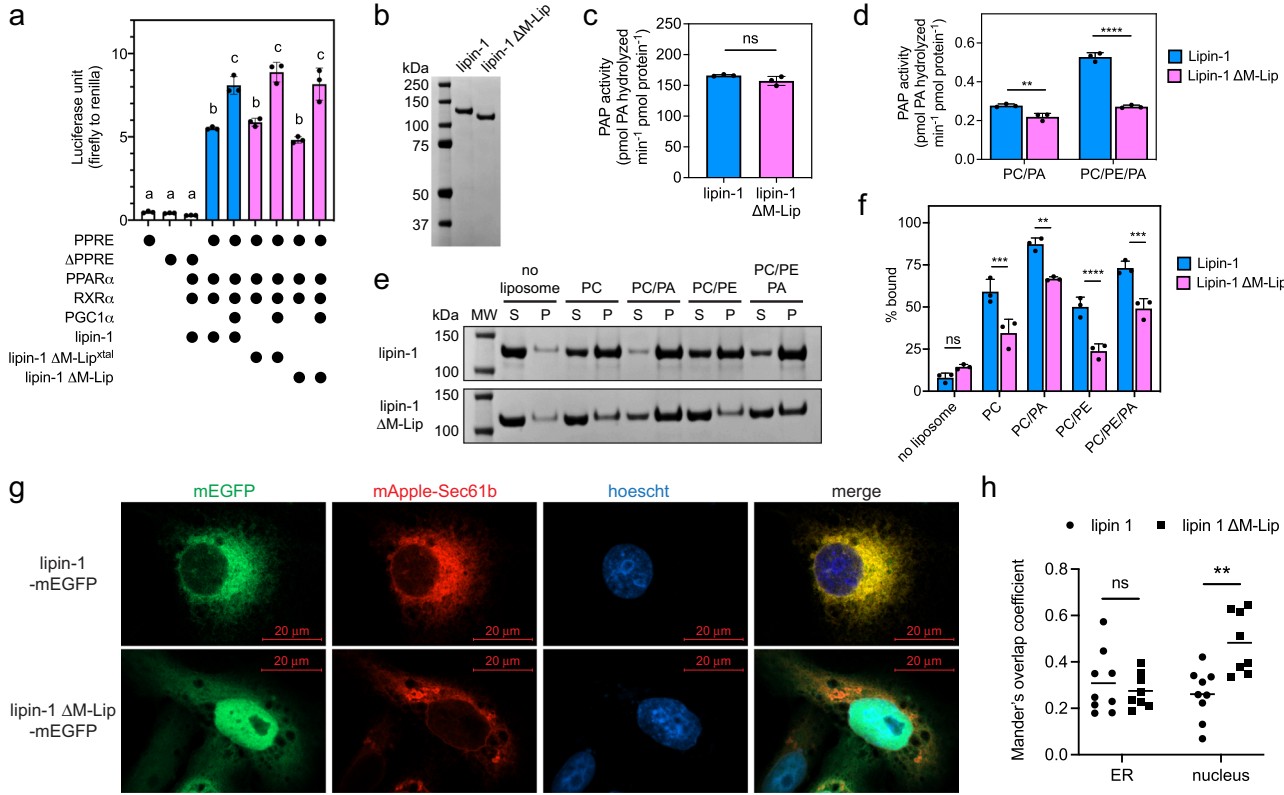

**Fig. 3 The M-Lip domain reduces lipin 1 membrane association and PAP activity. a** Deletion of the M-Lip or M-Lip^xtal domain does not affect lipin 1 transcriptional co-activation. HEK293 cells were co-transfected with lipin 1 or lipin 1 M-Lip domain deletion constructs with the transcription factors PPARα and RXRα, and the transcriptional co-activator PGC1α. Transcriptional co-activation was quantitated by measuring firefly luciferase activity, which is under the control of three peroxisome-proliferation response elements (PPREs). Firefly luciferase activity was normalized to Renilla luciferase from a control plasmid that was included in each transfection. Data are presented as the mean values ± SDs. $n = 3$ independent experiments. 1-way ANOVA with correction for multiple comparisons as determined by Tukey HSD post-hoc test. Letters (a, b, c) denote groups that are not statistically significant within their group ($p > 0.05$) and statistically significant to other groups ($p < 0.0001$). **b** SDS-PAGE of purified lipin 1 and lipin 1 ΔM-Lip used in PAP activity and liposome sedimentation assays. Purifications independently replicated $n = 3$ with similar results. **c** PAP activity of lipin 1 and lipin 1 ΔM-Lip in Triton X-100 mixed micelles with 5 mol% NBD-PA. Data are presented as mean values ± SDs. $n = 3$ independent experiments. Statistical analysis by a paired two-tailed $t$ test. ns, $p = 0.1554$. **d** PAP activity of lipin 1 and lipin 1 ΔM-Lip in 90 mol% PC or 70/20 mol% PC/PE liposomes with 10 mol% NBD-PA. PC phosphatidylcholine, PE phosphatidylethanolamine, PA phosphatidic acid. Data are presented as mean values ± SDs of $n = 3$ independent experiments analyzed by 2-way ANOVA with Tukey's multiple comparison. **$p = 0.0074$; ****$p < 0.0001$. **e** SDS-PAGE analysis of a liposome sedimentation assay reveals lipin 1 preferentially associates with liposomes containing 20 mol% PA. Deletion of the M-Lip domain (lipin 1 ΔM-Lip) results in a reduction of membrane association. S supernatant, P pellet, MW molecular weight markers. Representative images from $n = 3$ independent experiments. **f** Quantification of liposome association for lipin 1 and lipin 1 ΔM-Lip. Data are presented as mean values ± SDs. $n = 3$ independent experiments. Statistical analysis by 2-way ANOVA with Tukey's multiple comparison. ns, $p = 0.8424$; **$p = 0.0020$; ***$p < 0.001$; ****$p < 0.0001$. **g** Confocal microscopy images of Cos-7 cells transiently transfected with monomeric enhanced GFP (mEGFP) fusions of either lipin 1 or lipin 1 ΔM-Lip (green) and the ER marker mApple-Sec61b (red). Hoechst stain (blue), nucleus. Scale bar: 20 μm. $n = 1$ with independent images ($n = 5$) shown in Supplementary Fig. 9. **h** Mander's overlap coefficient calculated using JacoP plugin in ImageJ analyzed by 2-way ANOVA with Tukey's multiple comparison. $n = 9$ cells. ns, $p = 0.9342$; **$p = 0.0024$. Source data are provided as a Source data file.

**Deletion of the M-Lip domain reduces lipin PAP activity.** We next tested whether deletion of the M-Lip domain affected lipin 1 PAP activity in vitro using purified protein (Fig. 3b) and the fluorescent substrate nitrobenzoxadiazole-phosphatidic acid (NBD-PA). Wild-type and ΔM-Lip lipin 1 had near identical PAP activities when the substrate NBD-PA was incorporated into Triton X-100 mixed micelles (Fig. 3c, Supplementary Fig. 7).

We next assessed PAP activity with NBD-PA incorporated into liposomes composed of palmitoyl-oleoyl-phospholipids, which represent a more physiologically relevant in vitro system. Liposomes composed solely of phosphatidylcholine (PC) or a mixture of PC and phosphatidylethanolamine (PE) were used, as the membrane lipid PE has previously been shown to increase PAP activity of lipin 1[22,28]. The PAP activity of wild-type and ΔM-Lip lipin 1 were similar in PC liposomes (Fig. 3d). However,

unlike wild-type lipin 1, the activity of ΔM-Lip lipin 1 did not increase with the addition of PE (Fig. 3d). Thus, PAP activity in PC/PE liposomes was ~50% lower for ΔM-Lip lipin 1 compared to wild-type, which suggests that the M-Lip domain is involved in the PE-mediated effects on lipin 1 PAP activity.

**Deletion of M-Lip reduces membrane association and alters subcellular localization.** Given our findings that M-Lip reduces PAP activity on PC/PE liposomes, we hypothesized that the M-Lip domain affects lipin 1 binding to membranes. Using a liposome sedimentation assay, we found that lipin 1 bound to liposomes containing the neutral lipids PC or PC/PE, and liposome association was further enhanced by the presence of 20 mol% PA (Fig. 3e, f). Deletion of the M-Lip domain resulted in a consistent reduction of liposome association regardless of the liposome lipid

composition (Fig. 3e, f). As a negative control, lipin 1 sedimentation was not affected by lipid-induced aggregation (Supplementary Fig. 8), as previously observed for PAP from rat liver[35,36].

To assess if the M-Lip domain affects the subcellular distribution of lipin 1, we transiently transfected Cos-7 cells with wild-type and ΔM-Lip lipin 1 fused at their C-terminus with monomeric enhanced GFP (mEGFP) and assessed their subcellular localization using confocal microscopy. Lipin 1 co-localized with the ER marker mApple-Sec61b (Fig. 3g, h, Supplementary Fig. 9a). In contrast, ΔM-Lip lipin 1 also co-localized with the ER but accumulated in the nucleus (Fig. 3g, h, Supplementary Fig. 9b). We concluded that the M-Lip domain is necessary for proper subcellular localization and full-membrane binding in vitro.

**The isolated M-Lip domain binds membranes**. To verify that the M-Lip domain directly binds membranes, we conducted liposome sedimentation assays using the isolated M-Lip and M-Lip[xtal] domains purified from *Escherichia coli*. The M-Lip domain bound to PC liposomes, and membrane association was enhanced by the presence of the anionic lipids PA, phosphatidylserine, and phosphatidylinositol (Fig. 4a, b). In contrast, the M-Lip[xtal] domain, which lacks the conserved C-terminal hydrophobic and basic residues, exhibited weak association with liposomes that was not significantly affected by the presence of anionic lipids (Fig. 4a, b).

Next, we employed HDX-MS to identify the regions of the M-Lip domain that interact with membranes. HDX-MS experiments were conducted in the presence and absence of PC/PA/PE liposomes. PE was included as we and others have observed differences in lipin PAP activity in the presence of PE[22,37]. Several peptides were protected from deuterium exchange in the presence of liposomes. The strongest protection was observed for peptides containing a WWF motif (Fig. 4c, Supplementary Fig. 1), which are part of the conserved hydrophobic and basic residues at the C-terminal end of the M-Lip (Fig. 2a). Overall, HDX changes observed in the M-lip domain using the M-lip domain alone versus full-length lipin 1 were very similar with slight differences potentially due to variation in membrane binding parameters. Two additional peptides present in the crystallized M-Lip[xtal] domain were also protected (Fig. 4c). These peptides mapped to the same surface, which suggests that the core of the M-Lip[xtal] domain can also interact with membranes but is not the major determinant of membrane binding (Fig. 4c).

To determine if the isolated M-Lip domain localized to membranes within cells, Cos-7 cells were transfected with M-Lip fusions with mEGFP. A mEGFP M-Lip fusion strongly co-localized with the ER marker mApple-Sec61b (Fig. 4d, e, Supplementary Fig. 10a). In contrast, the mEGFP M-Lip[xtal] fusion accumulated in the nucleus with a minor co-localization with the ER, suggesting a role for the hydrophobic and basic residues at the C-terminus of M-Lip in membrane targeting (Fig. 4d, e, Supplementary Fig. 9b). Taken together, these results identify the M-Lip domain as a new type of membrane-binding domain.

**M-Lip role in adipogenesis**. Lastly, we sought to characterize the role of the M-Lip domain in lipin 1 function in adipocytes. *Lpin1* is expressed in a bi-phasic manner during adipocyte differentiation, and has specific roles in both the early stages of adipogenesis and in the formation of mature, lipid-laden adipocytes[5]. In pre-adipocytes, lipin 1 PAP activity regulates PA signaling to promote expression of a key adipogenic transcription factor, peroxisome proliferator-activated receptor γ (PPARγ)[5,38]. Lipin 1 PAP activity is also required in mature adipocytes for triglyceride synthesis and lipid hydrolysis[38,39]. We investigated whether the

M-Lip domain is required for the optimal activity of lipin 1 in the regulation of gene expression and lipid accumulation during adipocyte differentiation.

3T3-L1 preadipocytes were transfected with expression vectors for wild-type or ΔM-Lip lipin 1. We titrated expression levels of wild-type and ΔM-Lip lipin 1 to ensure that they were expressed at comparable levels (Supplementary Fig. 11a). Expression of adipocyte genes was monitored at intervals during differentiation, and neutral lipid accumulation was examined after 5 days (Fig. 5a). Wild-type lipin 1 expression, but not ΔM-Lip lipin 1 expression, increased the levels of oil red O-stained lipids beyond those observed in cells transfected with a vector control when assessed at day 5 (Fig. 5b and Supplementary Fig. 11b). Both wild-type and ΔM-Lip lipin 1 expression led to enhanced *Pparg* expression compared to vector controls, but ΔM-Lip was less effective than wild-type lipin 1 in promoting expression of the transcription factor CCAAT/enhancer binding protein α (*Cebpa*), or mature adipocyte genes such as fatty acid binding protein 4 (*Fabp4*) and adiponectin (*Adipoq*) (Fig. 5c). An independent replicate experiment also showed that ΔM-Lip lipin 1 was less effective at inducing lipogenic genes encoding acetyl-CoA carboxylase (*Acaca*) and diacylglycerol acyltransferase 1 (*Dgat1*) (Supplementary Fig. 11c).

## Discussion

This study identifies the M-Lip domain as a new protein fold that dimerizes and binds membranes. The M-Lip domain is functionally important for lipin subcellular localization, and enhancing adipogenesis, and can directly affect PAP activity in vitro in a manner that is dependent on membrane lipid content. With the exception of lipin 1 transcriptional co-activator function, which was unaffected, deletion of the M-Lip had a negative effect but did not completely abrogate any lipin function. This is consistent with the sufficiency of the N-Lip and C-Lip regions for lipin PAP activity[18] and all known disease mutations residing within the N-Lip or C-Lip regions.

We could identify the M-Lip domain in lipin homologs from several evolutionarily distant organisms, but not in *Sc* Pah1, nor in any non-lipin proteins from any species. Thus, the M-Lip domain appears to represent a unique feature of lipins and an evolutionary branch point that differentiates lipin PAP enzymes from mammals, invertebrates, ciliates, and plants from *Sc* Pah1.

Membrane association is the main regulatory mechanism that controls lipin PAP activity. Our HDX-MS studies reveal that membrane binding involves regions distributed throughout lipin 1. We propose mammalian lipins associate with membranes through a series of multi-valent interactions involving the N-terminal amphipathic helix, nuclear localization signal, Ig-like domain, HAD-like phosphatase domain, and the M-Lip domain. In this model, the M-Lip domain contributes one site for membrane binding and simultaneously doubles the number of membrane binding interactions through dimerization (Fig. 6).

All of the individual membrane-binding regions in lipins that have been characterized to date are responsive to the presence of anionic lipids, in particular PA[18,22,26,27,37]. This suggests that lipins may be recruited to cellular membranes in response to elevated PA levels. Given that PA is also the lipin enzyme substrate, this may reflect a mechanism to maintain low levels of PA. A role for PE may also exist, as PE has been hypothesized to impact the electrostatic charge of PA[22,37]. While we observed increased PAP activity in the presence of PE, intriguingly PE modestly decreased lipin 1 membrane association, which has also been observed for *Sc* Pah1[40]. In addition, the effects of acyl-chain composition and membrane fluidity on lipin activity warrants future study. This hypothesis derives from the similar specific

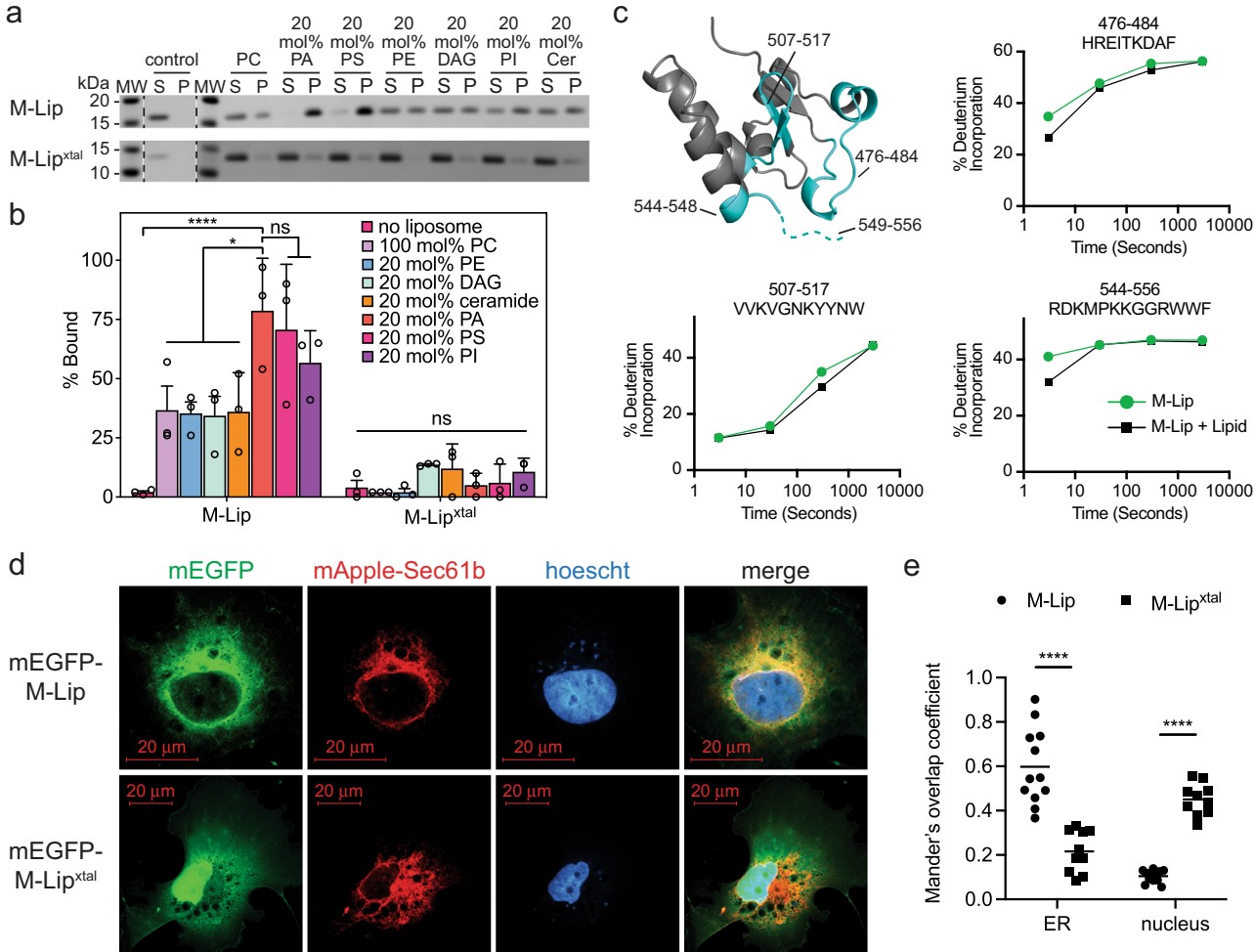

**Fig. 4 The M-Lip domain binds membranes in vitro and in cells. a** SDS-PAGE analysis of liposome sedimentation assays reveal the M-Lip domain binds membranes and preferentially associates with liposomes containing anionic lipids (PA, PS, and PI). The M-Lip[xtal] domain binds weakly to membranes. S supernatant, P pellet, MW molecular weight markers, PC phosphatidylcholine, PE phosphatidylethanolamine, PA phosphatidic acid, PS phosphatidylserine, DAG diacylglycerol, PI phosphatidylinositol, Cer ceramide. Representative images from $n = 3$ independent experiments. **b** Quantification of liposome association for the M-Lip and M-Lip[xtal] domains. Data are presented as mean values ± SDs. $n = 3$ independent experiments. Statistical analysis by 2-way ANOVA with Tukey's multiple comparison. ns, $p > 0.05$; *$p < 0.05$; ****$p < 0.0001$. **c** Regions that showed significant decreases in exchange (defined as >4%, >0.4 Da, and a two-tailed student $t$-test $p < 0.01$) in the presence of liposomes are colored in blue according to the legend and displayed on a single subunit of the lipin 1 M-Lip[xtal] domain. % deuterium incorporation of selected peptides at various time points (3, 30, 300, and 3000 s) in the absence and presence of liposomes. Data are presented as mean values ± SDs. $n = 3$ independent experiments. Most SDs are smaller than the size of the point. **d** Confocal microscopy images of Cos-7 cells transiently transfected with monomeric enhanced GFP (mEGFP) fusions of either the M-Lip or M-Lip[xtal] domains (green) and the ER marker mApple-Sec61b (red). Hoechst stain (blue), nucleus. Scale bar: 20 μm. $n = 1$ with independent images ($n = 3$) shown in Supplementary Fig. 10. **e** Mander's overlap coefficient calculated using JacoP plugin in ImageJ analyzed by 2-way ANOVA with Tukey's multiple comparison. $n = 11$ cells. ****$p < 0.0001$. Source data are provided as a Source data file.

activities we observed in comparison to other groups using Triton X-100 mixed micelles[22,41] (Fig. 3c), which was diminished in liposomes containing palmitoyl-oleoyl phospholipids (Fig. 3d). In contrast, a previous report found no major differences in specific activity when comparing mixed micelles and liposomes composed of di-oleoyl phospholipids[22].

As revealed by HDX-MS, the N-Lip and C-Lip regions are predominantly ordered in solution with substantial secondary structure. This finding suggests the N-Lip and C-Lip co-fold to form a split Ig-like domain and the HAD-like catalytic domain observed in *Tt* Pah2[18]. This is consistent with previous findings that a fusion of the N-Lip and C-Lip regions in both mouse lipin 2[18] and *Sc* Pah1[32] is sufficient for PAP activity in vitro. Notably, while the association of the N-Lip and C-Lip regions with one another appears to be constitutive in vitro, we cannot rule out that their association is transient or regulated in cells.

Here we expand the domain architecture of mammalian lipins to include M-Lip as a third protein domain that forms a stable dimer. Therefore, mammalian lipins must, at a minimum, be dimeric with the respective N-Lip and C-Lip regions co-folding either in cis (from the same subunit, Fig. 6) or in trans (from different subunits). We note that if the N-Lip and C-Lip interaction occurs in trans, this creates the potential to form larger oligomers when 4 or more lipin subunits combine. Lipin homo- and hetero-oligomerization has been demonstrated previously[34,42], and the size exclusion profile of recombinant lipin 1 yielded two peaks consistent with oligomerization beyond a dimer. Lastly, our data suggest that the conserved dimer interface of the M-Lip may contribute to both homo and hetero-dimerization of mammalian lipins. This provides a foundation for more detailed experiments to unravel the functional consequences of lipin oligomerization in the regulation of phospholipid and triglyceride synthesis.

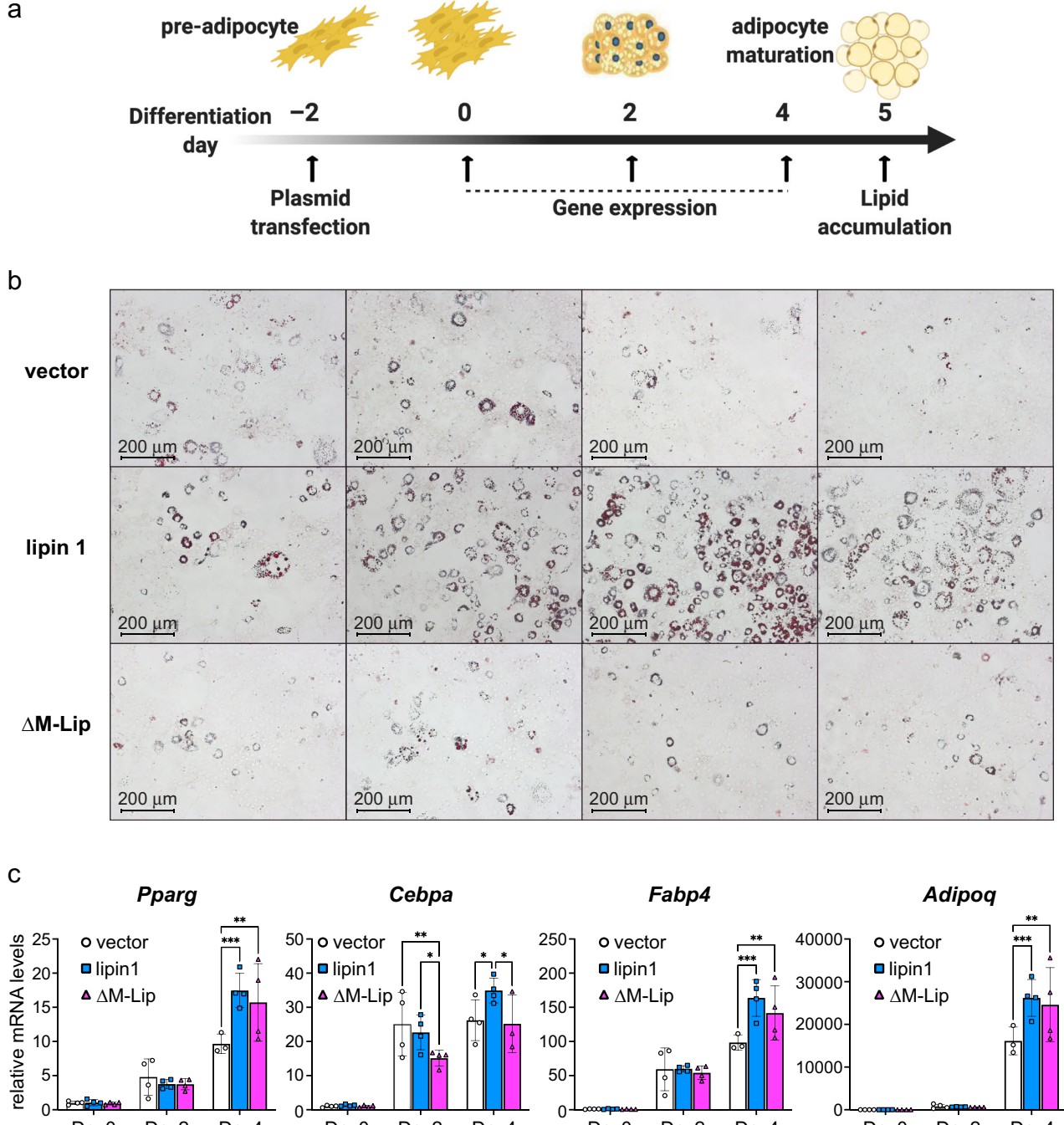

**Fig. 5 Optimal lipin 1 enhancement of adipogenesis requires the M-Lip domain. a** The ability of wild-type and ΔM-Lip lipin 1 to promote 3T3-L1 preadipocyte differentiation were studied by transfection with corresponding expression constructs followed by analysis of gene expression and lipid accumulation. **b** At day 5 after transfection, 3T3-L1 adipocytes were stained with oil red O to detect neutral lipid accumulation. Images shown are 100x magnification. Scale bar: 200 μm. $n = 4$ biologically independent experiments with independent replicate experiments shown in Supplementary Fig. 11b. **c** Expression of genes encoding adipogenic transcription factors PPARγ (*Pparg*) and C/EBPα (*Cebpa*), adipocyte fatty acid binding protein FABP4 (*Fabp4*), and the adipocyte hormone, adiponectin (*Adipoq*), at days 0, 2, and 4 of differentiation. Gene expression was analyzed by 2-way ANOVA, followed by paired *t*-tests when ANOVA results were significant. *$p < 0.05$; **$p < 0.01$; ***$p < 0.001$. Data are presented as mean values ± SD. $n = 4$ biologically independent experiments. Source data are provided as a Source data file.

## Methods

**Plasmids**. For expression in Sf9 insect cells, full-length mouse lipin 1 alpha (residues 1–891) was cloned into YM-Bac3 using SfoI and NotI restriction sites. YM-Bac3 is a modified version of pFastBac Htb (Invitrogen) that contains an N-terminal 6xHis tag followed by a Dual-Strep tag. The ΔM-Lip lipin 1 construct (residues 1–891 Δ458–565) was generated using the overlap extension method to delete the M-Lip domain and create an in-frame fusion between residues 457 and 566.

For *Escherichia coli* expression, the mouse lipin 1 (M-Lip domain, residues 458–565; M-Lip^xtal, residues 458–548) and mouse lipin 2 (M-Lip^xtal, residues 459–549) were cloned into pET28b using NdeI and NotI restriction sites. Mouse lipin 1 M-Lip^xtal was also cloned into ppSUMO with BamHI and NotI sites, which added a ULP1 cleavable N-terminal His-tagged SUMO fusion.

For expression in mammalian cells, full-length mouse lipin 1α constructs (WT, residues 1–891; ΔM-Lip, residues 1–891 Δ458–565; and ΔM-Lip^xtal, residues 1–891

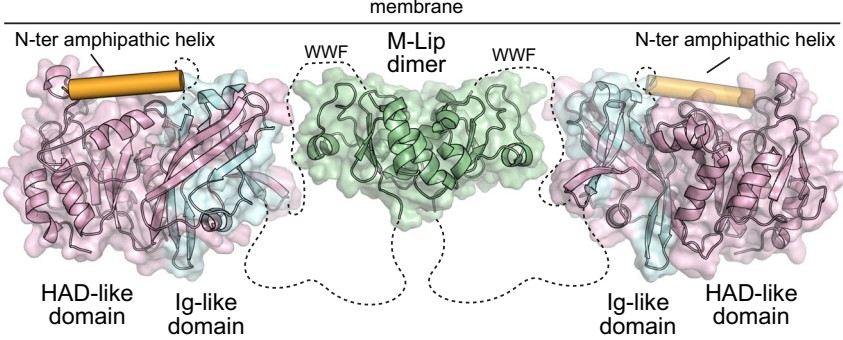

**Fig. 6 Structural model of mammalian lipin PAPs.** Proposed model for a dimeric lipin 1 protein with the N-Lip (blue) and C-Lip (pink) combining to form the Ig-like and HAD-like domains. Membrane association is driven by multi-valent interactions from the N-terminal amphipathic helix (orange), the active site of the HAD-like domain, regions of the Ig-like domain, and the dimeric M-Lip that contains a hydrophobic WWF motif that is flanked by basic residues.

Δ458–548) were cloned into pcDNA3.1 using EcoRI and NotI restriction sites in frame with a C-terminal V5-His tag. ΔM-Lip, and ΔM-Lip$^{xtal}$ deletion constructs were generated using the overlap extension method. The mouse lipin 1 M-Lip (mEGFP M-Lip) and M-Lip$^{xtal}$ (mEGFP M-Lip$^{xtal}$) domain fusions with monomeric enhanced GFP (mEGFP) located at the N-terminus were gene synthesized (BioBasic, Canada) and inserted into pcDNA3.1. Full-length mouse lipin 1 and lipin 1 ΔM-Lip fusions with mEGFP were generated by PCR using AgeI and PmeI restriction sites in pcDNA3.1. The primers for plasmid generation are listed in Supplementary Table 1. All plasmids were directly sequenced.

**Protein expression and purification**
*Full-length mouse lipin 1 and ΔM-Lip lipin 1 proteins.* Full-length mouse lipin 1 and ΔM-Lip lipin 1 were expressed in Sf9 cells using baculovirus. 300 mL of cells were infected with 1.0 mL of baculovirus at 3 million cells/mL at >95% viability and harvested 72 h later. Cell pellets were lysed in 40 mM Tris, pH 8.0, 300 mM NaCl, 10 mM 2-mercaptoethanol by sonication, and the lysates were centrifuged at 81,770 × *g* for 30 min. Lipin 1 and lipin 1 ΔM-Lip proteins were purified using Ni-NTA resin (GoldBio). Eluted protein was applied to Streptactin-XT resin equilibrated with equilibration buffer (100 mM Tris, pH 8.0, 150 mM NaCl, 10 mM 2-mercaptoethanol), then washed with equilibration buffer, and eluted with equilibration buffer containing 50 mM biotin. Proteins were further purified by size exclusion chromatography using a Superdex 200 10/300 column in 20 mM Tris, pH 8.0, 150 mM NaCl, and 10 mM 2-mercaptoethanol. Fractions containing lipin 1 and ΔM-Lip lipin 1 proteins were concentrated, flash-frozen, and stored at −80 °C.

*Mouse lipin 1 M-Lip domain.* The mouse lipin 1 M-Lip domain plasmid in pET28a was expressed in *Escherichia coli* BL21 (DE3) RIPL cells. Cells were grown at 37 °C in Ultra-High Yield Flasks (Thompson Instrument Company) to an OD$_{600nm}$ of 1.5 and then cooled at 10 °C for 2 h. Protein expression was induced with isopropyl β-D-1-thiogalactopyranoside (IPTG) at 15 °C for 12 h, cells were harvested by centrifugation and stored at −80 °C. Frozen cells were resuspended in buffer A (50 mM Tris-HCl, 60 mM imidazole, 500 mM NaCl, 5%(v/v) glycerol, 1% v/v Triton X-100, pH 7.4), lysed by sonication, centrifuged at 58,540 × *g* at 4 °C for 1 h, and applied to a gravity column with pre-equilibrated Ni-NTA resin (GoldBio). The column was washed with buffer A without Triton X-100, and protein was eluted with buffer B (50 mM Tris-HCl, 300 mM imidazole, 500 mM NaCl, 5% (v/v) glycerol, pH 7.4). The protein was diluted 4 fold in buffer C (50 mM HEPES, 50 mM NaCl, pH 7.35), applied to a HiTrap SP HP cation exchange column, washed with 5 column volumes of buffer C, and eluted with a linear gradient with buffer C supplemented with 1 M NaCl. Protein was aliquoted, flash frozen, and stored at −80 °C.

*Mouse lipin 1 and mouse lipin 2 M-Lip$^{xtal}$ domains.* The M-Lip$^{xtal}$ domains of mouse lipin 1 and mouse lipin 2 in pET28a were overexpressed in *Escherichia coli* BL21 (DE3) RIPL cells. Cells were grown at 37 °C in Ultra-High Yield Flasks (Thompson Instrument Company) to an OD$_{600nm}$ of 1.5 and then cooled at 10 °C for 2 h. Protein expression was induced with isopropyl β-D-1-thiogalactopyranoside (IPTG) at 15 °C for 18 h, cells were harvested by centrifugation, and stored at −80 °C. Frozen cells were resuspended in buffer A (50 mM Tris-HCl, 60 mM imidazole, 500 mM NaCl, 5%(v/v) glycerol, pH 7.4), lysed by sonication, centrifuged at 58,540 × *g* at 4 °C for 1 h, and applied to a gravity column with pre-equilibrated Ni-NTA resin (GoldBio). The column was washed with buffer A and protein was eluted with buffer B (50 mM Tris-HCl, 300 mM imidazole, 500 mM NaCl, 5% (v/v) glycerol, pH 7.4). The eluted protein was applied to a HiLoad Superdex 75 26/600 column (GE Healthcare) equilibrated with SEC Buffer (50 mM Tris-HCl, 150 mM NaCl, 5% (v/v) glycerol, pH 7.4). Pooled fractions from SEC were concentrated to 10-20 mg/ml, aliquoted, flash frozen, and stored at −80 °C.

*Mouse lipin 1 M-Lip$^{xtal}$ domain in ppSUMO.* The mouse lipin 1 M-Lip$^{xtal}$ domain that was crystallized was expressed and purified from the construct cloned into the ppSUMO plasmid. Expression conditions and the Ni-NTA purification protocol were identical as that described here for the M-Lip$^{xtal}$ domain in pET28a. However, after elution from the Ni-NTA column, the protein was digested with purified ULP-1 overnight at 4 °C. The digestion mixture was diluted 10 fold in buffer A, and re-applied to a Ni-column to remove the His-SUMO fusion. After, the M-Lip$^{xtal}$ protein was further purified by SEC using a HiLoad Superdex 75 26/600 column (GE Healthcare) equilibrated with SEC Buffer. Pooled fractions from SEC were concentrated to 10-20 mg/ml, aliquoted, flash frozen, and stored at −80 °C.

*Se-Met derivatized mouse lipin 1 M-Lip$^{xtal}$ protein.* The mouse lipin 1 M-Lip$^{xtal}$ domain derivatized with selenomethionine (Se-Met) was produced using B834 (DE3) cells. B834 cells were grown in M9 minimal medium supplemented with amino acids, Se-Met, and Kao and Michayluk Vitamin Solution (Sigma). Purification was identical to the M-Lip$^{xtal}$ domain.

**Liposome generation.** Large unilamellar vesicle (LUV) liposomes were prepared by the lipid extrusion method. Briefly, lipids dissolved in chloroform were dried under nitrogen gas for 30 minutes. For assays with full-length lipin 1 proteins, the liposomes were resuspended in 50 mM Tris, pH 7.5, 100 mM NaCl, 10 mM 2-mercaptoethanol buffer and LUVs were generated by 7 freeze-thaw cycles. Liposomes were composed of 20 mol% POPA (1-palmitoyl-2-oleoyl-sn-glycero-3-phosphate, Avanti Polar Lipids, #840857) and 0 or 40 mol% POPE (1-palmitoyl-2-oleoyl-sn-glycero-3-phosphoethanolamine, Avanti Polar Lipids, #850757) with the remaining 80 or 40 mol% as POPC (1-palmitoyl-2-oleoyl-glycero-3-phosphocholine, Avanti Polar Lipids, #850457).

For assays with the isolated M-Lip domains, liposomes were resuspended in PBS buffer. Followed by 3 rounds of freeze-thaw cycles with agitation, and extrusion with 100 nm size polycarbonate membrane filters (Avestin). Liposomes were composed of 73 mol% POPC, 5 mol% cholesterol (Avanti Polar Lipids, #700000P), 2 mol% PECF (1,2-dioleoyl-sn-glycero-3-phosphoethanolamine-N-(carboxyfluorescein), Avanti Polar Lipids, #810332) as visual guide and 20 mol% lipid of interest: POPA, POPS (1-palmitoyl-2-oleoyl-sn-glycero-3-phospho-L-serine, Avanti Polar Lipids, #840034), PODAG (1-palmitoyl-2-oleoyl-sn-glycerol, Avanti Polar Lipids, #800815), POPI (1-palmitoyl-2-oleoyl-sn-glycero-3-phosphoinositol, Avanti Polar Lipids, #850142) or ceramide (Avanti Polar Lipids, #8601518).

**Liposome sedimentation assays.** For the full-length lipin 1 and ΔM-Lip lipin 1 proteins, 20 µL of LUV liposomes in 50 mM Tris, pH 7.5, 100 mM NaCl, 10 mM 2-mercaptoethanol buffer were mixed with 20 µL of proteins to give a final concentration of 1.0 mM liposomes and 1.0 µM protein. For the M-Lip and M-Lip$^{xtal}$ proteins, 50 µL of LUV liposomes in pH 7.4 PBS buffer were mixed with 50 µL of proteins in PBS buffer, giving a final concentration of 1 mM liposomes and 50 µM protein. Reaction mixtures were incubated for 30 minutes and centrifuged at 100,000 × *g* at 4 °C for 1 h using a TLA100 fixed angle rotor (Beckman). The supernatant fraction was carefully removed, and the protein content of the pellet and supernatant fractions were analyzed by SDS-PAGE. All binding assays were performed at least three times and SDS-PAGE gel bands were quantified using ImageJ[43].

**BSA-lipid sedimentation assays.** To generate lipid-BSA complexes, 20 mol% POPA, 40 mol% POPE with the remaining of 80, 60, or 40 mol% as POPC were applied. Lipids in chloroform were dried under nitrogen gas, then resuspended with 10 mg/ml fatty acid free BSA solution in 50 mM Tris pH 7.5, 0.1 M NaCl, 10 mM beta-mercaptoethanol to a final concentration of 2 mM total lipid. The suspension solution was sonicated in water bath for 5 min per cycle for a total of 6

cycles, then centrifuged at $9168 \times g$ for 30 min. The supernatant was collected as lipid-BSA complexes. 20 μL lipid-BSA complexes were incubated with 20 μL lipin 1 at 4 °C for 30 min, then centrifuged at $100,000 \times g$ for 1 h. The supernatant was collected carefully from the tube. The pellet (P) and the supernatant (S) fractions were analyzed by SDS-PAGE.

**HDX-MS to determine ordered and disordered regions of full length lipin.** HDX reactions were conducted in 20 μL reaction volumes with a final concentration of 0.63 μM full length lipin 1 per sample. Exchange was carried out in triplicate for a single time point (3 s on ice). All tips and tubes were pre-chilled at 4 °C. Hydrogen deuterium exchange was initiated by the addition of 18 μL of ice-cold D2O buffer solution (20 mM HEPES pH 7, 100 mM NaCl) to the protein solution, to give a final concentration of 84.8% D2O. Exchange was terminated by the addition of 50 μL acidic quench buffer, giving a final concentration 0.6 M guanidine-HCl and 0.9% formic acid (pH ~2.5). Samples were immediately frozen in liquid nitrogen at −80 °C. A denatured protein sample for back exchange correction was generated by first denaturing the protein in 6 M guanidine for 1 h at 18 °C. Following denaturation, 18 μL of D2O buffer was added to the denatured protein and allowed to exchange for 15 min at 18 °C before quenching as described above. Samples were flash frozen in liquid nitrogen and stored at −80 °C until injection onto an ultra-performance liquid chromatography system for proteolytic cleavage, peptide separation, and injection onto a QTOF for mass analysis.

**HDX-MS mapping of full-length lipin 1 with liposomes.** Liposomes were made by resuspending the lipid film (20% POPA, 80% POPC) in lipid buffer (50 mM Tris pH 7.5, 0.1 M NaCl, 10 mM 2-mercaptoethanol) resulting in a final liposome concentration of 2 mM. The resuspended lipid was sonicated for 10 min, freeze-thawed seven times, and then extruded through a prewet 100 nm filter unit 10× times prior to usage in the experiment. HDX reactions were conducted in a final reaction volume of 20 μL with a lipin 1 protein concentration of 0.63 μM. Prior to the addition of deuterated solvent, 1 μL of lipin 1 was allowed to incubate with either 2 μL of 2 mM liposomes or 2 μL of the corresponding lipid buffer (protein:lipid ratio of 1:317). After the two-minute incubation period, 17 μL D2O buffer (20 mM HEPES pH 7, 100 mM NaCl, 0.5 mM TCEP, 94% D2O) was added with a final %D2O of 80.1% (v/v). The reaction was allowed to proceed for 3 s, 30 s, 300 s, or 3000 s at 18 °C before being quenched with 50 μL diluted ice-cold acidic quench buffer to a final pH of ~2.5 (as described above). All conditions and timepoints were created and run in triplicate. Samples were flash frozen in liquid nitrogen and stored at −80 °C until injection onto an ultra-performance liquid chromatography system for proteolytic cleavage, peptide separation, and injection onto a QTOF for mass analysis.

**HDX-MS mapping of the M-Lip domain with liposomes.** Liposomes were made by resuspending the lipid film (20% POPA, 60% POPC, 20% POPE) in lipid buffer (20 mM HEPES pH 7.5, 100 mM NaCl) resulting in a final liposome concentration of 4 mM. The resuspended lipid was sonicated for 10 min, freeze-thawed three times, and then extruded through a prewet 100 nm filter unit 11× times prior to usage in the experiment. HDX reactions were conducted in a final reaction volume of 20 μL with an M-Lip concentration of 1.2 μM. Prior to the addition of deuterated solvent, 1 μL M-Lip protein was allowed to incubate with either 1 μL of 4 mM lipid vesicles or 1 μL of the corresponding lipid buffer (protein:lipid ratio of 1:167). After a 2 min incubation period, 18 μL D2O buffer (20 mM HEPES pH 7.5, 100 mM NaCl, 94% D2O (v/v)) was added with a final %D2O of 84.9% (v/v). The reaction allowed to proceed for 3 s, 30 s, 300 s, or 3000 s at 18 °C before being quenched with 50 μL diluted ice-cold acidic quench buffer resulting in a final concentration of 0.6 M guanidine-HCl and 0.9% FA post quench to a final pH of ~2.5 (as described above). All conditions and timepoints were created and run in triplicate. Samples were flash frozen in liquid nitrogen and stored at −80 °C until injection onto an ultra-performance liquid chromatography system for proteolytic cleavage, peptide separation, and injection onto a QTOF for mass analysis.

**Protein digestion and MS/MS data collection.** Protein samples were rapidly thawed and injected onto an integrated fluidics system containing a HDx-3 PAL liquid handling robot and climate-controlled chromatography system (LEAP Technologies), a Dionex Ultimate 3000 UHPLC system, as well as an Impact HD QTOF Mass spectrometer (Bruker). The protein was run over either one immobilized pepsin column at 10 °C (order-disorder and full-length lipin) or two (M-Lip experiment) immobilized pepsin columns at 2 °C and 10 °C (Applied Biosystems; Poroszyme Immobilized Pepsin Cartridge, 2.1 mm × 30 mm; Thermo-Fisher 2-3131-00; Trajan; ProDx protease column, 2.1 mm × 30 mm PDX.PP01-F32) at 200 μL/min for 3 min. When using two pepsin columns, the protein is first run over the column in the 10 °C box, followed by running over the 2 °C column. The resulting peptides were collected and desalted on a C18 trap column at 2 °C (Acquity UPLC BEH C18 1.7 μm column (2.1 mm × 5 mm); Waters 186003975). The trap was subsequently eluted in line with a C18 reverse-phase separation column (2 °C) (Acquity 1.7 μm particle, 100 mm × 1 mm$^2$ C18 UPLC column, Waters 186002352), using a gradient of 3–35% B (Buffer A 0.1% formic acid; Buffer B 100% acetonitrile) over 11 min immediately followed by a gradient of 35–80% over 5 min. A blank method is run after every MS method to limit carryover, using

a 3–35% B gradient over 4 min followed by a gradient of 35–80% over 2 min. Mass spectrometry experiments acquired over a mass range from 150 to 2200 $m/z$ using an electrospray ionization source operated at a temperature of 200 °C and a spray voltage of 4.5 kV.

**Peptide identification.** Peptides were identified using data-dependent acquisition following tandem MS/MS experiments (0.5 s precursor scan from 150 to 2000 $m/z$; twelve 0.25 s fragment scans from 150 to 2000 $m/z$). MS/MS datasets were analyzed using PEAKS7 (PEAKS), and peptide identification was carried out by using a false discovery-based approach, with a threshold set to 1% using a database of known contaminants found in Sf9 and *Escherichia coli* cells[44]. The search parameters were set with a precursor tolerance of 20 ppm, fragment mass error 0.02 Da, charge states from 1 to 8, and allowing for phosphorylation at Serine, Threonine, and Tyrosine.

**Mass analysis of peptide centroids and measurement of deuterium incorporation.** HDExaminer Software (Sierra Analytics) was used to automatically calculate the level of deuterium incorporation into each peptide. All peptides were manually inspected for correct charge state and presence of overlapping peptides. Deuteration levels were calculated using the centroid of the experimental isotope clusters. The results for the experiment comparing lipin with and without liposomes are presented as relative levels of deuterium incorporation, with no control for back exchange, and the only correction was for the level of deuterium present in the buffer (either 84.9% or 80.1%). The fully deuterated sample in the order-disorder experiment allowed for a back-exchange correction in this specific experiment during digestion and separation. Differences in exchange in a peptide were considered significant if they met all three of the following criteria: >4% change in exchange, >0.4 Da difference in exchange, and a $p$-value <0.01 using a two-tailed Student's $t$-test. We arrived at the %D and #D criteria partially based on the average error of the HDX experiments, with the %D value being at least 5-fold greater than the average error. All compared samples were set within the same HDX experiment. The raw HDX data are shown in four different formats. The raw peptide deuterium incorporation graphs for a selection of peptides with significant differences are shown, with the raw data for all analyzed peptides in the source data. To allow for visualization of differences across all peptides, we utilized number of deuteron difference (#D) plots. These plots show the total difference in deuterium incorporation over the entire H/D exchange time course, with each point indicating a single peptide. We also generated #D plots showing the difference in deuterium incorporation for each individual timepoint. Butterfly plots were used to visualize the % deuterium incorporation of each individual peptide at each timepoint. The data analysis statistics for all HDX-MS experiments are in Supplementary Table 2 according to the guidelines of Masson et al.[29]. The mass spectrometry proteomics data have been deposited to the ProteomeXchange Consortium via the PRIDE partner repository[45] with the dataset identifier PXD022172.

**Crystallization and data collection.** All crystals were grown using the hanging-drop method by mixing 1.5 μL of reservoir solution mixed with 1.5 μL of protein solution at RT. Crystals of native and Se-Met mouse lipin 1 M-Lip$^{xtal}$ domain were grown in 5–15% PEG 3000, 0.1 M Tris-HCl, pH 7.0 by seeding with microcrystals generated using a Seed Bead Kit (Hampton Research, HR4-781). Crystal form 2 of mouse lipin 1 M-Lip$^{xtal}$ were grown in 0.2 M zinc acetate, 1 M NaCl, 0.1 M Imidazole, pH 8.0 by seeding with microcrystals. Crystals of mouse lipin 2 M-Lip$^{xtal}$ were grown in 0.2 M Calcium Acetate, 20% PEG 3350. All crystals were cryo-protected with 25% v/v glycerol and flash frozen in liquid nitrogen prior to data collection. Diffraction data for mouse lipin 1 native M-Lip$^{xtal}$ domain was collected at 0.979 Å at the APS NE-CAT 24-ID-C beamline (Table 1). Data for the Se-Met and crystal form 2 of the mouse lipin 1 M-Lip$^{xtal}$ domain were collected at 0.979 Å and 1.28 Å at the NSLS-II FMX 17-ID beamline at Brookhaven National Laboratory (Table 1). Diffraction data for mouse lipin 2 M-Lip$^{xtal}$ domain was collected at 0.979 Å at the Advanced Photon Source GM/CA ID-23B beamline at Argonne National Laboratory (Table 1).

**Data processing, structure determination, and refinement.** Diffraction data were integrated and scaled using XDS-DIALS[46] in CCP4[47]. Se-Met SAD phases were calculated to 2.3 Å resolution using Phenix AutoSol[48], which generated an initial map. The resulting map showed clear electron density for side-chains and the model was further improved by manual model building using COOT[49]. This yielded a nearly complete search model for molecular replacement with the 1.5 Å resolution native dataset. Subsequent model adjustments were carried out in COOT with further refinement with Phenix[50]. The finalized mouse lipin 1 M-Lip$^{xtal}$ domain structure was used as a search model for molecular replacement for crystal form 2 of mouse lipin 1 and the mouse lipin 2 M-Lip$^{xtal}$ domain using Phaser[51] in Phenix[50]. The final models were generated by manual model building in COOT and refinement in Phenix. Data collection and refinement statistics are provided in Table 1.

**SEC-MALS.** 1.6 mg of the mouse lipin 1 M-Lip$^{xtal}$ domain was injected onto a Superdex 200 Increase 10/300 (GE health) size exclusion column with constant

flow rate of 0.35 ml/min. Light scattering and refractive index data were collected by miniDAWN TREOS (Wyatt technology) and Optiab T-rEX (Wyatt technology), respectively. The refractive index change was measured differentially by mini-DAWN TREOS with a laser at a wavelength of 658 nm, and UV absorbance was measured with the diode array detector at 280 nm. Data were subsequently processed by ASTRA software 7 (Wyatt technology) and peak alignment and band broadening correction between the UV, MALS, and RI detectors were performed using Astra software algorithms.

**Cell culture and transfection**. Murine Hepa1-6 hepatoma cells (American Type Culture Collection #CRL-1830, Manassas, VA) and human HEK293 embryonic kidney cells (CRL-1573) were maintained in minimal essential medium (MEM) with 10% fetal bovine serum (FBS) (Corning Inc., Corning, NY). All experiments were performed with cells having 2–6 passages. Cells were transfected with plasmids using BioT reagent (Bioland Scientific, Paramount, CA).

**Co-Immunoprecipitation**. Co-immunoprecipitation experiments were performed to detect lipin 1 dimer formation. To unambiguously identify the interaction between two lipin monomers, equal numbers of Hepa1-6 cells were transfected with independent lipin 1 constructs that were fused to different epitopes (V5 or BirA*-HA). To detect lipin heterodimer formation, lipin 1 constructs with V5 epitope (wild-type lipin 1-V5, ΔMLip-V5, or ΔM-Lip$^{xtal}$-V5) were co-transfected with lipin 1-BirA*-HA, lipin 2-BirA*-HA or lipin 3-BirA*-HA. Two days after transfection, cells were harvested and lysed in 0.1% NP-40 with phosphatase inhibitor cocktails (Sigma-Aldrich, St. Louis, MO). After brief sonication, the supernatants were collected by centrifugation at $13,400 \times g$ for 10 min at 4 °C. 10% of the resulting protein was reserved as "input" and the remainder was immunoprecipitated. For immunoprecipitation of V5-tagged proteins, cell lysates were incubated with a 1:500 dilution of anti-V5 antibody (ThermoFisher Scientific, R960-25, Waltham, MA) at 4 °C overnight. Cell lysate/antibody mixture were then incubated with protein A/G-agarose beads (Santa Cruz Biotechnology, Inc., Santa Cruz, CA) for 2 hr at 4 °C. The precipitates were washed three times with lysis buffer, boiled in sample buffer for 5 min and subjected to immunoblot assay with a 1:1000 dilution of anti-HA antibody (Cell Signaling Technology, Inc., #3724, Danvers, MA) or 1:5000 dilution of anti-V5 antibody. Densitometric quantitation was performed via a ChemiDox XRS+ using the manufacturer's software (Bio-Rad, Hercules, CA).

Independent co-immunoprecipitation experiments were performed with lipin proteins having different epitope tags. For these experiments, full-length or ΔM-Lip lipin 1 tagged with V5 were immunoprecipitated with lipin 1-FLAG or lipin 3-Myc using a 1:500 dilution of anti-V5 antibody. When cells were harvested, 10% of protein was reserved as "input" and the rest was subjected to immunoprecipitation as described above. Immunoblots were detected with a 1:500 dilution of anti-FLAG antibody (#PA1-984B, Invitrogen, Carlsbad, CA) or a 1:1000 dilution of anti-Myc antibody (#2278, Cell Signaling, Danvers, MA). Gels were designed to allow direct comparison of the levels of protein input, flow-through, and immunoprecipitation of full-length and ΔM-Lip lipin 1. Densitometric quantitation was performed via a ChemiDox XRS+ using the manufacturer's software (Bio-Rad, Hercules, CA).

**Lipin 1 transcriptional coactivation assay**. Ability of wild-type and ΔM-Lip lipin 1 to coactivate PGC1α was assessed according to previous methods[8,15]. HEK293 cells were transfected with a PPRE-firefly luciferase reporter plasmid together with phRL-TK *Renilla* luciferase control vector (Promega, Madison, WI), pCMX-PPARα, pCMX-PGC1α, pCMX-RXR, and wild-type lipin 1, ΔMLip-V5, or lipin 1 ΔM-Lip$^{xtal}$-V5 expression constructs. Two days after transfection, cells were collected and luciferase activity determined with the Dual-Luciferase Assay System (Promega, Madison, WI).

**3T3-L1 adipogenesis**. 3T3-L1 preadipocytes were cultured and differentiated according to previous methods[52]. To investigate the function of the M-Lip domain in adipogenesis, cells were reverse-transfected at differentiation day –2 (confluent) with pcDNA, wild-type lipin 1-V5, or ΔM-Lip-V5 (BioT reagent, Bioland Scientific). At day 0, cells were changed to differentiation medium (ZenBio, DM2L1500), and at day 2 were switched to maintenance medium (ZenBio, AM-1-L1). RNA isolation was performed at day 0, day 2, and day 4 of differentiation. Cell morphology was monitored by brightfield phase-contrast and Oil red O staining.

**RNA extraction and quantitative RT-PCR**. Total RNA was isolated from 3T3-L1 cells using TRIzol (Life Technologies) and reverse transcribed (iScript, Bio-Rad). Real-time RT-PCR was performed with a Bio-Rad CFX Connect Real-Time PCR Detection System using SsoAdvanced SYBR Green Supermix and *Tbp* mRNA as a normalization control. Primers were as follows:
*Pparg* (AACTCTGGGAGATTCTCCTGTTGA, TGGTAATTTCTTGTGAAG TGCTCATA);
*Cebpa* (GAACAGCAACGAGTAACCGGGTA, GCCATGGCCTTGACCAAG GAG);
*Fabp4* (AACCTGGAAGCTTGTCTCCA, CACGCCCAGTTTGAAGGAAA);
*Adipoq* (TGTTCCTCTTAATCCTGCCCA, CCAACCTGCACAAGTTCCCTT);

*Acaca* (GCCTCTTCCTGACAAACGAG, TGACTGCCGAAACATCTCTG);
*Fasn* (GTTGGCCCAGAACTCCTGTA, GTCGTCTGCCTCCAGAGC);
*Dgat1* (CTGAATTGGTGTGTGGTGATG, AGGGGTCCTTCAGAAACAG AG);
*Tbp* (ACCCTTCACCAATGACTCCTATG, ATGATGACTGCAGCAAAT CGC).

**Oil red O stain**. At day 5 of adipocyte differentiation, cells were fixed in 10% formalin for 20 min, washed with 60% isopropanol, and stained with oil red O solution (0.2% w/v in isopropanol). Cell images were captured at 100x magnification. Pictures from 4 independent wells per treatment group were used.

**PAP activity assays**. Micelles containing 5 mol% nitrobenzoxadiazole-phosphatidic acid (NBD-PA) (Avanti Polar Lipids, #9000341) and Triton X-100 (Research Products International Corp, #400001) were generated in a buffer containing 50 mM Tris pH 7.5, 100 mM NaCl, 10 mM 2-mercaptoethanol, and 4 mM MgCl$_2$, to give a final bulk concentration of 80 μM NBD-PA. Liposomes used for PAP assays were composed of 10 mol% NBD-PA with 0 mol% or 40 mol% POPE with the remaining as POPC (90 mol% or 50 mol%) and prepared in buffer containing 50 mM Tris, pH 7.5, 100 mM NaCl, 10 mM 2-mercaptoethanol, and 4 mM MgCl$_2$ to give a final bulk concentration of 150 μM NBD-PA.

In a 50 μL reaction mixture, 45 μL NBD-PA mixed micelles or liposomes were incubated with 5 μL lipin 1 or lipin 1 ΔM-Lip (total 5 ng protein) for 30 min at 30 °C. Reactions were quenched by the addition of 150 μL of CHCl$_3$/MeOH (1:1), then vortexed and centrifuged at $771 \times g$ for 3 min. The organic phase was collected, dried under N$_2$(g), and resuspended with 100 μL methanol containing 0.2% formic acid and 1 mM ammonium formate.

Both the mixed-micelle and liposome samples were analyzed by HPLC using a Spectra 3 μm C8SR column (3 μm particle, 150 × 3.0 mm, Peeke Scientific, #S-3C8SR-FJ) under the following conditions: solvent A: water containing 0.2% formic acid and 1 mM ammonium formate; solvent B: methanol containing 0.2% formic acid and 1 mM ammonium formate; flow rate: 0.5 ml/min. The gradient profile started at 80% for solvent B and was increased to 98% B after 7 min, then kept at 98% for 3 min. 10 μL samples were injected onto the column, which was kept at 35 °C at all runs. The fluorescent signal was detected at excitation and emission wavelengths of 470 and 530 nm, respectively. The detector signal was recorded and integrated by using Agilent technology OpenLAB CDS ChemStation edition software.

**Confocal microscopy**. COS-7 cells (American Type Culture Collection, CRL-1651) were seeded in 6-well plate with a cover glass inside and were transfected using lipofectamine 2000 (Thermo Fisher, #11668027) with mouse mEGFP M-Lip or mEGFP M-Lip$^{xtal}$ and mApple-Sec61b, or lipin 1 mEGFP or ΔM-Lip lipin 1 mEGFP in pcDNA3.1. 48 hr after transfection cells were rinsed with PBS three times, then fixed with 4% paraformaldehyde containing 5 μg/ml Hoechst 33342 stain (Thermo Fisher, #62249) for 10 min at room temperature, then rinsed three times with PBS. The cover glass was then mounted onto a microslide. The cells were visualized using Zeiss Axio imager M2 confocal microscope. Co-localization was quantified using a Mander's overlap coefficient calculated using the JacoP plugin in ImageJ[53].

**Reporting summary**. Further information on research design is available in the Nature Research Reporting Summary linked to this article.

## Data availability

The coordinates and structure factors in this study have been deposited in the Protein Data Bank under accession codes 7KIH, 7KIL, and 7KIQ. The mass spectrometry proteomics data in this study have been deposited in the ProteomeXchange Consortium via the PRIDE partner repository[45] with the dataset identifier PXD022172. Source data are provided with this paper.

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

## Acknowledgements

We thank the staff at the NSLS-II (AMX and FMX) and APS (GM-CAT and NE-CAT) beamlines for assistance during data collection, and Vitaly Citovsky and Benoit Lacroix (Stony Brook) for access to their confocal microscope supported by the NIH grant 5R01GM05022423. This work was supported by the National Institutes of Health grants R35 GM128666 (M.V.A.), P01 HL090553 (K.R.), and P01 HL028481 (K.R.), the American Heart Association grants 17SDG33410860 (M.V.A.) and 18POST34060200 (H.W.), the NSERC Discovery Grant NSERC-2020-04241 (J.E.B., the Michael Smith Foundation for Health Research (J.E.B., Scholar Award 17686), and a Stony Brook URECA award (N.M.P.).

## Author contributions

W.G. performed all experiments with the isolated mouse lipin 1 M-Lip domain including protein purifications, crystallization experiments, SEC-MALS, and liposome sedimentation assays. J.W.Y. purified and crystallized the mouse lipin 2 M-Lip^xtal domain. W.G. and M.V.A. determined and refined the final crystal structures. SG purified full-length lipin 1 and ΔM-Lip proteins and performed PAP assays, liposome sedimentation assays, and confocal microscopy experiments. K.D.F. and R.M.H. performed the HDX-MS experiments for lipin 1 and the M-Lip, respectively. K.D.F., R.M.H., and J.E.B. analyzed all HDX-MS data.

H.W. performed co-immunoprecipitation, transcriptional co-activator, and adipogenesis studies. Y.M.C. conceived of and generated the YM-Bac3 plasmid. N.M.P. generated key constructs for experiments. W.G., S.G., H.W., K.R., J.E.B., and M.V.A. contributed intellectual and strategic input. K.R., J.E.B., and M.V.A. supervised work and provided funding support. W.G., S.G., K.R., and M.V.A. drafted the initial manuscript with contributions from K.D.F., J.E.B., and H.W. W.G., S.G., K.R., J.E.B., K.R., and M.V.A. edited the final manuscript. All authors approved the final manuscript.

## Competing interests

The authors declare no competing interests.
