## [Peer Review File · Nature Communications]

REVIEWER COMMENTS

Reviewer #1 (Remarks to the Author):

Gu et al. provide a relevant and novel structural and functional insights on lipin 1, a Mg-dependent phosphatidic acid phosphatase involved in several biological functions related to lipid metabolism, transportation and storage. The authors characterized how lipins bind membranes and identified a new middle lipin domain, conserved in mammalian lipins, which plays a crucial role for the enzyme function. The manuscript is well written and, mostly the conclusions are adequately supported by the data. There are a few areas that the authors need to address prior to publication, particularly focused on HDX-MS

Results

Line 102: change in "...and their deuterium content quantified"

Line 103-104: "Residues with no MS coverage were all located between the N-Lip and C-Lip regions, and near the highly basic nuclear localization signal" As described in the introduction of this manuscript, the linker region can be modified with several PTMs. Are those present in the region of the linker lacking sequence coverage? If so, the lack of sequence coverage could be due to those PTMs, which might have occurred in the expression system. Have the authors tried to add them as a variable modifications in their database search? If not, I suggest checking for that.

Line 106-107: What about the experiment on ice described in the method session? Nothing is mentioned about it in the results session, and I cannot see any data plotted in figures or reported in the supplementary table.

Line 131-132: References might be missing.

Fig. 1b: Are the D% normalized according to the fully labelled control? Normalization with the fully labelled control would be the most appropriate way to conduct such analysis. In the supplementary table, uptake values for the time point (3 sec) and the fully labelled control are presented. I can see in the supplementary data set, that for some regions the uptake of the fully labelled control is at around 30%, which indicates a very high BE or, perhaps, that the region is not fully exchanged (please, see my comment on the method session on how the fully labelled control has been conducted). This makes the data a bit difficult to be interpreted. If authors are confident on the fact that the protein is fully labelled at this condition, deuterium uptake should be normalized with the value of the BE control and coloured according to that % on the figure. Additionally, the figure could be improved by reformatting the x-axis. The major unit should be less than 200 (please, restrict to 50 or so) to help the reader better orientating along the sequence.

Fig. 1c: I suggest colouring a small part of the linker in blue and indicate that this segment is region 544-555.

Fig. 2c. There is a marker in the x-axis of peptide 680-685, which is only present in this plot. Please, remove it, unless this has a particular meaning that I missed.

Fig. 1e: Very difficult to read, everything is overlapped. I suggest moving this graph to the supplementary info and make it bigger and clearer. Please, also add two lines indicating the threshold of significance in term of absolute uptake. In general, a better representation is a graph showing the differences in deuterium uptake at every time point for every peptide (for instance: something similar to the typical butterfly plots returned by DynamX), and not a differential sum. Unless, authors have not considered a differential sum (it is not very clear how the thresholds are calculated; see following comment).

Line 201: Fig. 2a, h \rightarrow wrong figure indication; it is Fig. 2a, g

Line 204: Fig. 2i \rightarrow wrong figure indication; it is Fig. 2h

Fig. 3b is not referred anywhere in the text

Line 263: What do you mean for “unbiased manner”?

Line 264: “HDX-MS experiments were conducted in the presence and absence of PC/PA liposomes.” In the method session, it is written that you used liposomes made of POPA, POPC and POPE. Here POPE is not indicated. Is it present or not? If yes, can authors comment on why, for this experiment, they added POPE in the liposome composition? Was not better to add PS, for instance, which showed a greater association to M-lip and a marked decrease with M-lipxtal? I am referring to what I see in fig. 4 a,b. Also, authors clearly highlighted that anionic phospholipids enhance membrane association. Therefore, I am confused on why authors picked up POPE, which is zwitterionic and showed a weak association. Can authors elaborate on that?

HDX-MS experiment assessing how M-lip binds to the membrane. Can authors speculate on a potential explanation for not having identified significant effects in peptides spanning residues 507-517 and 476-484 with the full-length Lip? This does not seem to be due to the protein:lipid ratio used (which actually favours more binding in the experiment with the full-length compared to the experiment with M-lip). Can this effect be specific for POPE? Please, elaborate.

Fig. 4c. Peptides showing effects seem not to be correctly indicated on the structure. Please correct the figure and check if there is also a mistake in colouring. Please, add a dashed line indicating the non-crystallized C-terminal and colour part of it, to make the figure clearer.

Materials and Methods

Liposome generation: Have authors measured liposome diameter and polydispersity index of the liposome sample?

HDX-MS to determine ordered and disordered regions of full length lipin:

1. Was the labelling at 4 degree performed in a cold room or on a thermomixer? How the temperature was kept constant? Please explain. Also, in the supplementary table 1, there are two different temperature values reported (18 and 4 degrees). This appears incorrect to me.

2. How many microliters of quench buffer? I guess, 20ul? Final pH after quenching?

3. Usually, for the fully labelled control, chaotropic agents (deuterated Urea or deuterated Gnd-HCl) are also inserted in the labelling buffer. As you inserted it in the protein buffer only, the protein might refold at some extent when diluted in the labelling buffer and full deuteration not achieved in 15 minutes. Indeed, there are some regions showing a suspectable particularly high BE (reported in supplementary data set). This could be perhaps a signature of incomplete deuteration under this condition, due to the presence of a defined higher order structure. Additionally, the average BE and the IQR should be calculated and reported it in the supplementary table 1, as recommended in Masson et al.

HDX-MS mapping of full-length lipin 1 with liposomes

1. Please, write the protein: lipid ratio. If I am not wrong, this is 1:317.

2. Out of curiosity: why is TCEP added to the deuterated buffer? Is this to mimic the beta-mercaptoethanol present in the protein buffer? Is the final pH of the buffer 7? TCEP generally makes solution very acidic, and it would be useful to know if authors have checked the pH of the final solution (even though there is the buffer HEPES)

HDX-MS mapping of the M-Lip domain with liposomes

Please, write the protein: lipid ratio. If I am not wrong, this is 1:167. Is there a particular reason why authors have not used the same ratio as for the experiment with the full-length lipin?

Protein digestion and MS/MS data collection

1. Two protease columns: very interesting setup. For clarity, I suggest that authors briefly but better describe this setup here. Where the pepsin columns are placed? One outside and one inside the refrigerated chamber? Are two digestions carried out to enhance the enzymatic efficiency because they are both conducted at low temperature? Which experiments were run with two pepsin columns and why some needed double digestion? This should be specified

2. Please make your way to describe column dimensions uniform along the text. Sometimes there are two times mm, sometimes only once, sometimes only once with the (2). Please, choose only one way. I suggest "number mm x number mm"

3. At which temperature was the LC separation conducted? 0 degree? It is not written

4. Is 16 minutes the length of the whole run or the length of 5-36%? If the latter, this appears to me a very long gradient for an HDX-MS experiment (normally gradients are 7-9 minute long). Can authors comment on this?

5. Authors did not remove lipids prior the LC-MS analysis, and this does not surprise me as many lipids are very well tolerated at the LC-MS level, especially when injected in low amounts. However, authors should specify if they designed a particular wash method to let the lipids elute out the analytical column in between runs or if the run has a saw-tooth wash procedure or such after reaching 36% of B. This may justify the unusual length of the run!

Mass analysis of peptide centroids and measurement of deuterium incorporation

Statistical method: it is not very clear how the threshold values are calculated. Which statistical approach have been used to calculate 0.4 Da? And 5% and 4%? This is an important point that should be clearly addressed by the authors. Also, the table in supplementary reports 5% and 4% as threshold for the two different experiments. Here only 5% is mentioned. Can the authors correct or clarify?

Reviewer #2 (Remarks to the Author):

The manuscript by Gu et al examines the M-domain, a new domain within the located within the central region of the lipin family of phosphatidic acid phosphatases (PAP) that the authors hypothesize to promote both dimerization and membrane association. A portion of the M-domain is crystallized. Deuterium exchange shows that the presence of liposomes reduces labeling in the N-, M-, and C-LIP domains. Deletion of the M-LIP domain in the context of full-length Lipin-1 reduces membrane association and PAP activity, changes cellular localization and fails to promote adipogenesis, but does not affect transcriptional co-activation.

I really liked this manuscript as it provides several interesting observations about lipin structure and function. The work is well done and will be a nice contribution to the field of glycerolipid biology. A significant number of additional experiments has greatly added to the structural studies including examinations of the effect of the M-domain on transcriptional activity, cellular localization, PAP activity, and membrane binding. About my only generalized concern is that deletion of the M-Lip domain does not seem critical to any particular aspect of Lipin-1 function, that is, deletion has only modest effects on dimerization, PAP activity, and membrane association. Nevertheless, the authors are generally careful in their interpretation of the data. Overall this is a good manuscript that should be acceptable with minor revisions.

A few suggestions and minor criticisms;

1. Why were no peptides identified in the region between 120-256? Is there any logical region why there were none identified? This region has been covered extensively by LC-MS analysis in the past. The NLS would not be seen of course, but the SRD is of interest.
2. Are the enzymatic assays in Fig 3c&d linear with time and concentration of protein?
3. Fig. 3c&d_ Looks like deletion of the M-domain does not affect PAP activity towards PA except in the presence of PE. It has been hypothesized that PE impacts the electrostatic charge of PA but the liposome binding data in Fig 3f suggests that deletion of the M-Lip domain modestly reduces membrane association regardless of the phospholipid composition. How is this consistent? Further, under these conditions the apparent charge of PA seems irrelevant to lipin-1 membrane association (PC/PE/PA). How does this square with the increase in binding seen with anionic phospholipids in Fig 4a?
4. Some concern about non-specific sedimentation of Lipin-1 and the M-domain. The 'binding' of lipin-1 or the M-Lip domain to liposomes could be an artifact of aggregation-induced sedimentation, previously seen with PAP activity and fatty acids [PMID: 15820750, 11876265]. This concern would be alleviated by simple additional experiment solubilizing equivalent concentrations of phospholipids with BSA, mixing with Lipin-1 proteins, and centrifuging. Another concern with Fig 4a&b is the presence of TX100 during isolation and purification of the M-Lip vs M-Lipxtal proteins. Would be better if both were isolated under the same conditions (i.e., in the presence of 1% TX-100).
5. Fig. 2f, Please indicate approximate molecular weights of controls on x-axis.
6. It would be helpful if the authors provided statistical significance for biochemical experiments (Fig 2e-h, 3a-f, 4a-b), and quantitation and statistics for the cellular localization experiments (Fig 3g, 4d).
7. For Fig. 5b please demonstrate transfection efficiency with GFP (or GFP-tagged Lipin-1) and calculate efficiency. A macroscale visualization of the effect on Oil Red O accumulation would be useful as well, e.g., picture of entire plate. Also, what type of statistical analysis was used for calculation of significance for qPCR analysis?
8. The specific activity of Lipin-1 in TX100 and especially in liposomes (Fig 3&d) seems pretty low compared to previous reports [PMID: 20231281, 23426360]. Please address.

Typos or minor wording concerns

The authors use the word 'influences' throughout the manuscript. Influences is a very vague word and is not appropriate, please simply describe the effect (i.e., inhibits, reduces, etc).

Page 2, line 36, "...binds membranes through an N-terminal amphipathic helix and a middle lipin (M-Lip) domain...". Also show through C-Lip domain.

Page 7, line 204, the authors should acknowledge here that the reduction in co-immunoprecipitation is quite modest. Further, page 11, line 346, in the text it says, "...may mediate..", but was just pointed out other interactions may also mediate dimerization. Would instead might say '..may contribute..".

Page 7, line 204. No Fig 2i, refers to Fig 2h

Page 10, line 298, usual nomenclature is 'CCAAT/enhancer binding protein'

Page 11, line 326, would be best for clarity if all membrane binding domains referred to are actually listed.

Page 17, line 545, "...the following criteria: >5% change in exchange..", but in Supplemental Table 1 for lipin-1 and lipin-1 + membrane it says >4%. Also Fig 1 legend says >4%.

Reviewer #3 (Remarks to the Author):

The manuscript by Gu et. al. describes the molecular and structural mechanisms by which lipin phosphatidic acid phosphatases (PAPs) bind membranes and demonstrates this process is critical for the function of lipins.

In particular, the authors discovered and characterized a domain in the mammalian Lipins harboring a novel protein fold able to form dimers by combining HDX-MS and X-ray crystallography. They then demonstrate this domain is essential for the function of lipin in adipogenesis.

Overall the methodology and data are robust and I did not find any flaws that should prohibit its publication.

Gu et al – Response to reviewer comments

We thank the reviewers for their positive review of the manuscript and their constructive comments. We have addressed each comment and provide a point-by-point response below. Accompanying our revised submission is a word document with track changes included to help follow the changes.

Reviewer #1 (Remarks to the Author):

Gu et al. provide a relevant and novel structural and functional insights on lipin 1, a Mg-dependent phosphatidic acid phosphatase involved in several biological functions related to lipid metabolism, transportation and storage. The authors characterized how lipins bind membranes and identified a new middle lipin domain, conserved in mammalian lipins, which plays a crucial role for the enzyme function. The manuscript is well written and, mostly the conclusions are adequately supported by the data. There are a few areas that the authors need to address prior to publication, particularly focused on HDX-MS

Results

Line 102: change in "...and their deuterium content quantified"

We have changed this as suggested by the reviewer

Line 103-104: "Residues with no MS coverage were all located between the N-Lip and C-Lip regions, and near the highly basic nuclear localization signal" As described in the introduction of this manuscript, the linker region can be modified with several PTMs. Are those present in the region of the linker lacking sequence coverage? If so, the lack of sequence coverage could be due to those PTMs, which might have occurred in the expression system. Have the authors tried to add them as a variable modifications in their database search? If not, I suggest checking for that.

Thank you for the suggestion. In our original analysis, database searches were carried out using the sequence of the purified protein, with variable phosphorylation allowed at S, T, and Y. Even with these variables accounted for no peptides were identified in the stated regions between the N-Lip and C-lip region. To make our protocol clearer, we have now included a more detailed analysis of this in the methods section. Specifically, we have added the statement in the peptide identification section of the methods (new text is underlined). "The search parameters were set with a precursor tolerance of 20 ppm, fragment mass error 0.02 Da, charge states from 1-8, and allowing for phosphorylation at Serine, Threonine, and Tyrosine."

Line 106-107: What about the experiment on ice described in the method session? Nothing is mentioned about it in the results session, and I cannot see any data plotted in figures or reported in the supplementary table.

The experiment describing the pulse experiment were described in lines 466-470 originally (now 553-559). We apologise for the oversight in the inclusion of this data in the

supplementary table. The data was included in Fig 1b which is now stated in the figure legend. This is also shown in Supplementary Table 1.

Line 131-132: References might be missing.

Thank you. We have added a helical wheel diagram to Figure 1c, updated the legend, and added appropriate references.

Fig. 1b: Are the D% normalized according to the fully labelled control? Normalization with the fully labelled control would be the most appropriate way to conduct such analysis. In the supplementary table, uptake values for the time point (3 sec) and the fully labelled control are presented. I can see in the supplementary data set, that for some regions the uptake of the fully labelled control is at around 30%, which indicates a very high BE or, perhaps, that the region is not fully exchanged (please, see my comment on the method session on how the fully labelled control has been conducted). This makes the data a bit difficult to be interpreted. If authors are confident on the fact that the protein is fully labelled at this condition, deuterium uptake should be normalized with the value of the BE control and coloured according to that % on the figure. Additionally, the figure could be improved by reformatting the x-axis. The major unit should be less than 200 (please, restrict to 50 or so) to help the reader better orientating along the sequence.

We have edited the axis as suggested by the reviewer. The data shown in Fig. 1B has been normalised to the fully deuterated control experiment. A fully deuterated sample was generated empirically by using different times of exposure to the GdHCl solution. The experiment reported had the highest level of deuterium incorporation. There is a potential for some regions to not be fully deuterated, and as such we have changed the wording from fully deuterated, to denatured sample. However, our goal with this experiment was to determine the back exchange occurring in highly dynamic regions, allowing us to identify regions with either no or very unstable secondary structure.

Fig. 1c: I suggest colouring a small part of the liker in blue and indicate that this segment is region 544-555.

We have added the blue color to this part of the figure. We have also added a C-terminal linker with appropriate coloring as well.

Fig. 2c. There is a marker in the x-axes of peptide 680-685, which is only present in this plot. Please, remove it, unless this has a particular meaning that I missed.

We have removed the additional line in the x-axis.

Fig. 1e: Very difficult to read, everything is overlapped. I suggest moving this graph to the supplementary info and make it bigger and clearer. Please, also add two lines indicating the threshold of significance in term of absolute uptake. In general, a better representation is a graph showing the differences in deuterium uptake at every time point for every peptide (for instance: something similar to the typical butterfly plots returned by DynamX),

and not a differential sum. Unless, authors have not considered a differential sum (it is not very clear how the thresholds are calculated; see following comment).

The goal with this figure was to clearly show all differences in a single graph, in a limited space. To address the reviewers concerns we have added additional graphs in the supplement (supplementary figure 1) (covering Figure 1E, and 4C). This includes both a classic butterfly plot (with %D incorporation), as well as a #D difference plot, with a line that represents the data for each time point. Importantly, we do not use the data in Fig 1E to determine thresholds. This is described by the figure legend for Figure 1, where any peptide that had a change in deuterium incorporation meeting the following criteria at any timepoint was mapped on the structure (>4%, >0.4 Da, and a student t-test $p < 0.01$)

Line 201: Fig. 2a, h wrong figure indication; it is Fig. 2a, g
Thank you. We have corrected this error.

Line 204: Fig. 2i wrong figure indication; it is Fig. 2h
Thank you. We have corrected this error.

Fig. 3b is not referred anywhere in the text
We now refer to Fig. 3b in the sentence: "We next tested whether deletion of the M-Lip domain affected lipin 1 PAP activity in vitro using purified protein (Fig. 3b)"

Line 263: What do you mean for "unbiased manner"?

We have removed the unbiased manner statement.

Line 264: "HDX-MS experiments were conducted in the presence and absence of PC/PA liposomes." In the method session, it is written that you used liposomes made of POPA, POPC and POPE. Here POPE is not indicated. Is it present or not?

We apologise for the oversight as the liposome did contain PE as described in the methods. We have corrected this in text.

If yes, can authors comment on why, for this experiment, they added POPE in the liposome composition? Was not better to add PS, for instance, which showed a greater association to M-lip and a marked decrease with M-lipxtal?

I am referring to what I see in fig. 4 a,b. Also, authors clearly highlighted that anionic phospholipids enhance membrane association. Therefore, I am confused on why authors picked up POPE, which is zwitterionic and showed a weak association. Can authors elaborate on that?

PE was used since PE has previously been shown to influence the PAP activity of lipin, and we also observed a difference in PAP activity of full-length wild type lipin vs. Delta M-Lip. We have added this sentence to clarify our rationale: "PE was included as we and others have observed differences in lipin PAP activity in the presence of PE^{22, 37}."

HDX-MS experiment assessing how M-lip binds to the membrane. Can authors speculate on a potential explanation for not having identified significant effects in peptides spanning residues 507-517 and 476-484 with the full-length Lip? This does not seem to be due to the protein:lipid ratio used (which actually favours more binding in the experiment with the full-length compared to the experiment with M-lip). Can this effect be specific for POPE? Please, elaborate.

The differences in HDX seen in binding for the M lip domain were right at the borderline of significance. There likely will be altered binding to membranes in the full length construct (membrane occupancy time, k_{on} , k_{off} , etc) that could lead to subtle shifts explaining the very minor differences observed between full length and M-lip alone. We have added text describing possible mechanisms of difference between the two. However, this does not change our hypothesis on the mechanism of the M-lip domain in Lipin function. The text added is: "Overall, HDX changes observed in the M-lip domain using the M-lip domain alone versus full-length lipin 1 were very similar with slight differences potentially due to variation in membrane binding parameters."

Fig. 4c. Peptides showing effects seem not to be correctly indicated on the structure. Please correct the figure and check if there is also a mistake in colouring. Please, add a dashed line indicating the non-crystallized C-terminal and colour part of it, to make the figure clearer.

The figure has been modified. All coloring is correct.

Materials and Methods Liposome generation: Have authors measured liposome diameter and polydispersity index of the liposome sample?

We did not measure the diameter and polydispersity of these specific vesicles, however, we have used this exact protocol extensively to generate liposomes of similar composition for HDX-MS experiments. Previous experiments have revealed that the majority of particles had diameters ranging from 50-150 μM with this number of extrusions.

HDX-MS to determine ordered and disordered regions of full length lipin:

1. Was the labelling at 4 degree performed in a cold room or on a thermomixer? How the temperature was kept constant? Please explain. Also, in the supplementary table 1, there are two different temperature values reported (18 and 4 degrees). This appears incorrect to me.

We have expanded on the exact details of this experiment, with these details included in the methods. Tips utilised in the experiment were stored at 4°C until right before analysis. The experiment was performed in a temperature controlled 18°C room, with all samples and tubes maintained in an ice slurry solution. The denatured sample were allowed to equilibrate at 18°C. We have fixed all tables and methods to clearly describe this approach.

2. How many microliters of quench buffer? I guess, 20ul? Final pH after quenching?

We used 50 uL of quench, with these details now being included. The final pH was ~2.5. These details have been added.

3. Usually, for the fully labelled control, chaotropic agents (deuterated Urea or deuterated Gnd-HCl) are also inserted in the labelling buffer. As you inserted it in the protein buffer only, the protein might refold at some extent when diluted in the labelling buffer and full deuteration not achieved in 15 minutes. Indeed, there are some regions showing a suspectable particularly high BE (reported in supplementary data set). This could be perhaps a signature of incomplete deuteration under this condition, due to the presence of a defined higher order structure. Additionally, the average BE and the IQR should be calculated and reported in the supplementary table 1, as recommended in Masson et al.

We usually generate FD samples empirically, where we utilise different conditions, to obtain a condition leading to the maximum incorporation of deuterium. The reviewer is correct that there may still be some secondary structure that refolds even at high levels of denaturant. We have relabeled this experiment as the denatured control to indicate this uncertainty. We have reported the average BE and IQR as suggested by the reviewer.

HDX-MS mapping of full-length lipin 1 with liposomes

1. Please, write the protein: lipid ratio. If I am not wrong, this is 1:317.

We have included this value

2. Out of curiosity: why is TCEP added to the deuterated buffer? Is this to mimic the beta-mercaptoethanol present in the protein buffer? Is the final pH of the buffer 7? TCEP generally makes solution very acidic, and it would be useful to know if authors have checked the pH of the final solution (even though there is the buffer HEPES)

This was a typo, no TCEP was used in the buffer.

HDX-MS mapping of the M-Lip domain with liposomes

Please, write the protein: lipid ratio. If I am not wrong, this is 1:167. Is there a particular reason why authors have not used the same ratio as for the experiment with the full-length lipin?

We have included this value in the text. The ratio is slightly different because of the concentration of the protein stock used for these particular experiments.

Protein digestion and MS/MS data collection

1. Two protease columns: very interesting setup. For clarity, I suggest that authors briefly but better describe this setup here. Where the pepsin columns are placed? One outside and one inside the refrigerated chamber? Are two digestions carried out to enhance the enzymatic efficiency because they are both conducted at low temperature? Which experiments were run with two pepsin columns and why some needed double digestion? This should be specified

We have now expanded our description of this in the methods. Briefly, we have a pepsin column in the 10°C box in the LEAP PAL setup, but we have found greatly enhanced digestion of including a 2nd column in series located at 2°C in the cold box when using the Applied Biosystems Pepsin columns. This setup was used for the M-Lip experiment. The order disorder and the full-length lipin experiments were carried out using a Trajan pepsin column where one column at 10°C is sufficient for optimal digestion.

2. Please make your way to describe column dimensions uniform along the text. Sometimes there are two times mm, sometimes only once, sometimes only once with the (2). Please, choose only one way. I suggest “number mm x number mm”

We have fixed this uniformly in the methods.

3. At which temperature was the LC separation conducted? 0 degree? It is not written

We have added this detail in the methods (2°C).

4. Is 16 minutes the length of the whole run or the length of 5-36%? If the latter, this appears to me a very long gradient for an HDX-MS experiment (normally gradients are 7-9 minute long). Can authors comment on this?

We have described all of the LC methods in more depth in the methods. We use a slightly longer method (11 minutes), however, we still have back exchange values that are consistent with other HDX labs (~30% average).

5. Authors did not remove lipids prior the LC-MS analysis, and this does not surprise me as many lipids are very well tolerated at the LC-MS level, especially when injected in low amounts. However, authors should specify if they designed a particular wash method to let the lipids elute out the analytical column in between runs or if the run has a saw-tooth wash procedure or such after reaching 36% of B. This may justify the unusual length of the run!

We utilized a standard sawtooth blank method after every run (which we use for all samples, with lipid or not). These details have been added to the methods.

Mass analysis of peptide centroids and measurement of deuterium incorporation
Statistical method: it is not very clear how the threshold values are calculated. Which statistical approach have been used to calculate 0.4 Da? And 5% and 4%? This is an important point that should be clearly addressed by the authors. Also, the table in supplementary reports 5% and 4% as threshold for the two different experiments. Here only 5% is mentioned. Can the authors correct or clarify?

We use a three tiered system to describe significance (not just % and #D). This is based on (>4%, >0.4 Da, and a student t-test $p < 0.01$). The % and #D values were generated based on the average standard error observed, where we select a %D value at least

greater than 5 fold over the average. The %D and #D combined thresholds is used to not bias for very short peptides (%D) or very long peptides (#D). This is a well established protocol for determining statistical difference in HDX experiments. We have included more details of this in the methods. Both projects have been set to a %D threshold of 4% which has now been corrected in the text and the supplement.

Reviewer #2 (Remarks to the Author):

The manuscript by Gu et al examines the M-domain, a new domain within the located within the central region of the lipin family of phosphatidic acid phosphatases (PAP) that the authors hypothesize to promote both dimerization and membrane association. A portion of the M-domain is crystallized. Deuterium exchange shows that the presence of liposomes reduces labeling in the N-, M-, and C-LIP domains. Deletion of the M-LIP domain in the context of full-length Lipin-1 reduces membrane association and PAP activity, changes cellular localization and fails to promote adipogenesis, but does not affect transcriptional co-activation.

I really liked this manuscript as it provides several interesting observations about lipin structure and function. The work is well done and will be a nice contribution to the field of glycerolipid biology. A significant number of additional experiments has greatly added to the structural studies including examinations of the effect of the M-domain on transcriptional activity, cellular localization, PAP activity, and membrane binding. About my only generalized concern is that deletion of the M-Lip domain does not seem critical to any particular aspect of Lipin-1 function, that is, deletion has only modest effects on dimerization, PAP activity, and membrane association. Nevertheless, the authors are generally careful in their interpretation of the data. Overall this is a good manuscript that should be acceptable with minor revisions.

A few suggestions and minor criticisms;

1. Why were no peptides identified in the region between 120-256? Is there any logical region why there were none identified? This region has been covered extensively by LC-MS analysis in the past. The NLS would not be seen of course, but the SRD is of interest.

We were obviously very interested in mapping this region with MS to assess both local order/disorder and potential membrane interacting regions. Thus, when we did not observe peptides for these regions, we included database searches using the sequence of the purified protein with variable phosphorylation allowed at S, T, and Y. Even with these variables accounted for no peptides were identified in the stated regions between the N-Lip and C-lip region. This was done prior to our original submission. However, we did not mention in our original methods about including phosphorylation in our peptide identification protocol. We have now modified the methods section to include this. Specifically, we have added the statement in the peptide identification section of the methods (new text is underlined). "The search parameters were set with a precursor

tolerance of 20 ppm, fragment mass error 0.02 Da, charge states from 1-8, and allowing for phosphorylation at Serine, Threonine, and Tyrosine.”

2. Are the enzymatic assays in Fig 3c&d linear with time and concentration of protein?

Yes, the enzymatic assays are linear for both time and protein concentration. We have included this data in a new Supplementary Figure 6.

3. Fig. 3c&d_Looks like deletion of the M-domain does not affect PAP activity towards PA except in the presence of PE. It has been hypothesized that PE impacts the electrostatic charge of PA but the liposome binding data in Fig 3f suggests that deletion of the M-Lip domain modestly reduces membrane association regardless of the phospholipid composition. How is this consistent? Further, under these conditions the apparent charge of PA seems irrelevant to lipin-1 membrane association (PC/PE/PA). How does this square with the increase in binding seen with anionic phospholipids in Fig 4a?

We are aware of the previous studies that have demonstrated PE increases the PAP activity of lipin PAPs and membrane association in the presence of PA. For this reason, we included PE as a variable in our assays. As mentioned above, we had the same hypothesis-- that deletion of the M-Lip would decrease membrane binding in the presence of PE, given that it eliminates the PE-mediated activation of recombinant lipin-1. However, the results from our experiments did not follow this hypothesis. We agree that it is intriguing to see differential effects on PAP activity versus membrane binding. We do not have an obvious and clear explanation at this point. We can note that similar results have been reported for the yeast homolog *S. cerevisiae* Pah1 (Kwiatek and Carman, JLR 2020), where *Sc* Pah1 has higher activity in PC/PE/PA liposomes vs. PC/PA liposomes, but membrane association is higher in PC/PA liposomes vs. PC/PE/PA liposomes. We now expand our discussion with the following: “A role for PE may also exist, as PE has been hypothesized to impact the electrostatic charge of PA^{22, 39}. While we observed increased PAP activity in the presence of PE, intriguingly PE modestly decreased lipin 1 membrane association, which has also been observed for *Sc* Pah1⁴⁰.”

4. Some concern about non-specific sedimentation of Lipin-1 and the M-domain. The ‘binding’ of lipin-1 or the M-Lip domain to liposomes could be an artifact of aggregation-induced sedimentation, previously seen with PAP activity and fatty acids [PMID: 15820750, 11876265]. This concern would be alleviated by simple additional experiment solubilizing equivalent concentrations of phospholipids with BSA, mixing with Lipin-1 proteins, and centrifuging.

To address this first point, we have conducted the suggested experiments. The results show that addition of equivalent concentrations of phospholipids complexed with BSA does not result in recombinant lipin-1 pelleting after ultracentrifugation. Specifically, we see a small band of lipin-1 pellet in the absence of liposomes, BSA, or BSA/lipids, but the small band in the pellet does not increase in the presence of lipids. These results are now shown in supplementary figure 7, and the text has been modified accordingly. We state

that “These results were not affected by lipid-induced aggregation (Supplementary Figure 7), as previously observed for PAP from rat liver^{35, 36}.”

Another concern with Fig 4a&b is the presence of TX100 during isolation and purification of the M-Lip vs M-Lipxtal proteins. Would be better if both were isolated under the same conditions (i.e., in the presence of 1% TX-100).

Triton X-100 was included only for the M-Lip purification, and not for the M-Lip(xtal) purification, as without Triton X-100 the M-Lip would pellet during centrifugation after cell lysis. However, Triton X-100 was only included in the initial purification step and was not included in the wash or elution steps of the Ni-NTA affinity column, nor in the cationic exchange column to purify M-Lip. We acknowledge that some Triton X-100 molecules could remain bound to M-Lip after these purification steps, however we and others commonly include Triton X-100 at the lysis step to prevent membrane association of peripheral membrane proteins during cell lysis (see Khayyo et al, 2020). In the case that some Triton X-100 molecules remain bound to M-Lip, it is expected that the presence of this detergent would reduce membrane association, which is not the result we obtained. Thus, it is unclear to us how including Triton X-100 would affect the results of the liposome sedimentation assays reported in Fig. 4a and 4b. We have not made any changes to the manuscript to address this point and hope this explanation is satisfactory.

5. Fig. 2f, Please indicate approximate molecular weights of controls on x-axis.

We have included this data in Fig. 2f, as well as more detailed info in a new Supplementary Fig. 4.

6. It would be helpful if the authors provided statistical significance for biochemical experiments (Fig 2e-h, 3a-f, 4a-b), and quantitation and statistics for the cellular localization experiments (Fig 3g, 4d).

Where appropriate, we have now analyzed the data using statistical analyses and include this information in the figures and figure legends. A description of each panel is below.

2e- It is not clear to us how a statistical significance analysis would be useful for SEC-MALS data. MALS reports an absolute MW. We now include in the main text the MW and report the std dev for this calculation. 25.6 +/- 0.6 kDa

2f- We now include MW standards in Fig. 2f, and a new Supplementary Fig. 4.

2h- Completed and presented in supplemental figure 5 with additional co-immunoprecipitation experiments demonstrating the same result.

3a- Statistical analysis was via ANOVA corrected for multiple comparisons. Letter assignments showing groups that statistically differ from one another were by Tukey HSD post-hoc test.

3c – PAP assay in Triton X-100 mixed micelles. Student t test confirmed this was not significant.

3d – PAP assay in liposomes. 2-way ANOVA showed significant differences when comparing wild-type and delta M-Lip lipin in both PC ($p < 0.01$) and PC/PE ($p < 0.0001$)

liposomes. While the analysis suggested there was a difference in PC liposomes, this difference very small. We have chosen to keep the text the same, where we state the activities are “similar” in PC liposomes, as our errors bars are small for these experiments.

3f – Completed using 2-way ANOVA analysis.

4b – Completed using 2-way ANOVA analysis.

3g/4d - Co-localization data. We have now quantified the co-localization of our GFP-tagged lipin and M-Lip proteins with the ER and nucleus expressed using a Mander's overlap coefficient that was calculated using the JacoP plugin in ImageJ. Differences between groups were assessed using a 2-way ANOVA analysis. This has now been included in the methods section and the Mander's overlap coefficient included in the main figures.

7. For Fig. 5b please demonstrate transfection efficiency with GFP (or GFP-tagged Lipin-1) and calculate efficiency.

We have used 3T3-L1 transfection for many types of experiments across several years [for example, Phan et al. (2009) *Hum Mol Genet*, PMID 19124532; Peterfy et al. (2010) *J Biol Chem*, PMID 19955570; Chella Krishnan (2019) *Molec Metab*, PMID 31767179; Link et al. (2020) *J Clin Invest*, PMID 32701509], and no longer routinely check for expression efficiency. It is typically low in 3T3-L1 cells (10–20%), but the critical point is that all transfections within an experiment are performed in parallel with cells plated and harvested at the same time and transfected with the same amounts of plasmid.

A macroscale visualization of the effect on Oil Red O accumulation would be useful as well, e.g., picture of entire plate.

Although we agree that this is sometimes useful, the effects on adipocyte differentiation here are much more modest than, say, a gene knockout, and may not show up well with a macro view of plates. The gene expression data are a more sensitive and quantitative readout of differences in adipocyte differentiation throughout an entire plate of cells and are consistent with the oil red O fields. Furthermore, the oil red O results from multiple experiments are replicable.

Also, what type of statistical analysis was used for calculation of significance for qPCR analysis?

A 2-way ANOVA was used to calculate statistical significance of the qPCR analysis. Where ANOVA results were significant, it is permissible to perform paired t tests for specific comparisons, which are annotated by asterisks in the figures. This is now indicated in the figure legend.

8. The specific activity of Lipin-1 in TX100 and especially in liposomes (Fig 3&d) seems pretty low compared to previous reports [PMID: 20231281, 23426360]. Please address.

We appreciate this concern and have looked into it extensively.

For the mixed micelle assays, the specific activity of recombinant lipin in our hands seems comparable to the previous studies by the Carman (PMID: 20231281) and Harris (PMID: 23426360) groups, thus we believe we can rule out that the recombinant lipin we purified is less active and that the NBD-PA substrate we used causes any artifacts.

In our hands, the specific activity was 2.5 molecules of PA hydrolyzed per second per molecule of recombinant lipin (herein described as just 2.5/sec). Our assays used a bulk concentration of 80 μ M NBD-PA and 5 mol% NBD-PA. In comparison, the Carman group reported a k_{cat} of 58/sec using a much higher bulk concentration of PA (1mM). Looking at Fig. 8A where the effects of bulk PA concentration on activity are examined, the lower end of their data (\sim 80 μ M PA) has \sim 10 fold less activity than 1mM PA (we conservatively estimate this value at 3U/mg at \sim 80 μ M PA vs. the 30U/mg at 1mM PA). Thus, our specific activity appears similar to Carman (\sim 5.8/sec) at this lower concentration of PA. For the Harris group, they report specific activity in mixed micelles in Fig. 1C of their manuscript in the units: μ mol PA hydrolyzed/mg recombinant lipin. At 30 min, this value is \sim 10 μ mol/mg. Assuming a MW for lipin of 100,000 g/mol (for simple calculations), this converts to \sim 0.5/sec, which while slightly lower to our activity, is again comparable. The Harris group used a slightly higher but similar concentration of PA at 200 μ M PA and a slightly higher mol% at 9.1 mol% PA. Thus, we conclude from this that our recombinant lipin is active and that the NBD-PA has no detrimental effects on activity.

As mentioned by the reviewer, for the liposome activity we do observe significantly less specific activity in comparison to the Harris group. Comparing the experimental conditions, these are similar with regards to bulk concentration of PA (150 μ M by us vs. 200 μ M by Harris group) and mol% (10 mol% PA by both us and Harris group). The major difference between our assays is that we used PO phospholipids with one saturated (palmitoyl-) and one unsaturated acyl chain (oleoyl-), whereas the Harris group used DO phospholipids with two unsaturated acyl chains (di-oleoyl-). Given that we have similar activities to other groups with mixed micelles, we suspect that the difference in specific activity observed using liposomes is due to the differences in the saturation of the phospholipid acyl chains. This is certainly something that should be examined but believe is outside the scope of this current manuscript. Our rationale for choosing PO-based lipids was because PO-lipids better reflect the lipid composition of ER membranes, which typically do not have phospholipids with two unsaturated chains. In addition, DO-lipids, in particular PE, is known to form inverted micelles at high concentrations of DOPE ($>$ 70%), instead of liposomes with a bilayer.

To address this issue, we have now added the following to our discussion:

“In addition, the effects of acyl-chain composition and membrane fluidity on lipin activity warrants future study. This derives from the similar specific activity we observed in comparison to other groups^{22, 41} using Triton X-100 mixed micelles (Fig. 3c), which was diminished in liposomes containing palmitoyl-oleoyl phospholipids (Fig. 3d). In contrast, a previous report found no major differences in specific activity when comparing mixed micelles and liposomes composed of di-oleoyl phospholipids²².

Typos or minor wording concerns

The authors use the word 'influences' throughout the manuscript. Influences is a very vague word and is not appropriate, please simply describe the effect (i.e., inhibits, reduces, etc).

We appreciate this suggestion and agree. We have made several changes in the abstract and manuscript to clarify the effects of M-Lip deletion.

Page 2, line 36, "...binds membranes through an N-terminal amphipathic helix and a middle lipin (M-Lip) domain...". Also show through C-Lip domain.

Thank you. We now mention the "Ig-like domain and HAD phosphatase catalytic core" in the abstract, to be fully consistent with the HDX data.

Page 7, line 204, the authors should acknowledge here that the reduction in co-immunoprecipitation is quite modest.

We have now added "modestly" to this co-immunoprecipitation sentence. New text underlined: "Consistent with this hypothesis, deletion of the complete M-Lip domain or the M-Lip^{xtal} domain modestly reduced the ability of lipin 1 to co-immunoprecipitate with lipin 1, lipin 2 and lipin 3 (Fig. 2h)."

Further, page 11, line 346, in the text it says, "...may mediate..", but was just pointed out other interactions may also mediate dimerization. Would instead might say '..may contribute..'.
We now write: "Lastly, our data suggest that the conserved dimer interface of the M-Lip may contribute to both homo and hetero-dimerization of mammalian lipins."

We now write: "Lastly, our data suggest that the conserved dimer interface of the M-Lip may contribute to both homo and hetero-dimerization of mammalian lipins."

Page 7, line 204. No Fig 2i, refers to Fig 2h

This has been corrected.

Page 10, line 298, usual nomenclature is 'CCAAT/enhancer binding protein'

This has been corrected.

Page 11, line 326, would be best for clarity if all membrane binding domains referred to are actually listed.

Agreed. All membrane binding domains are now listed. The next text reads: "We propose mammalian lipins associate with membranes through a series of multi-valent interactions involving the N-terminal amphipathic helix, nuclear localization signal, Ig-like domain, HAD-like phosphatase domain, and the M-Lip domain. In this model, the M-Lip region

contributes one site for membrane binding and simultaneously doubles the number of membrane binding interactions through dimerization (Fig. 6)."

Page 17, line 545, "...the following criteria: >5% change in exchange..", but in Supplementary Table 1 for lipin-1 and lipin-1 + membrane it says >4%. Also Fig 1 legend says >4%.

Both projects have been set to a %D threshold of 4% which has now been corrected in the text and the supplement.

Reviewer #3 (Remarks to the Author):

The manuscript by Gu et. al. describes the molecular and structural mechanisms by which lipin phosphatidic acid phosphatases (PAPSs) bind membranes and demonstrates this process is critical for the function of lipins.

In particular, the authors discovered and characterized a domain in the mammalian Lipins harboring a novel protein fold able to form dimers by combining HDX-MS and X-ray crystallography. They then demonstrate this domain is essential for the function of lipin in adipogenesis.

Overall the methodology and data are robust and I did not find any flaws that should prohibit its publication.

Thank you for the positive feedback on our manuscript.

REVIEWERS' COMMENTS

Reviewer #1 (Remarks to the Author):

The authors have adequately addressed all comments.

Reviewer #2 (Remarks to the Author):

The authors have fully answered all criticisms from my initial review. This is a great manuscript and look forward to seeing it in publication.